# The Quantization Model of Neural Scaling

**Eric J. Michaud**,[*] **Ziming Liu, Uzay Girit, and Max Tegmark**
MIT & IAIFI

## Abstract

We propose the *Quantization Model* of neural scaling laws, explaining both the observed power law dropoff of loss with model and data size, and also the sudden emergence of new capabilities with scale. We derive this model from what we call the *Quantization Hypothesis*, where network knowledge and skills are "quantized" into discrete chunks (*quanta*). We show that when quanta are learned in order of decreasing use frequency, then a power law in use frequencies explains observed power law scaling of loss. We validate this prediction on toy datasets, then study how scaling curves decompose for large language models. Using language model gradients, we automatically decompose model behavior into a diverse set of skills (quanta). We tentatively find that the frequency at which these quanta are used in the training distribution roughly follows a power law corresponding with the empirical scaling exponent for language models, a prediction of our theory.[2]

## 1 Introduction

In the aggregate, larger neural networks trained on more data perform better than smaller neural networks trained on less data, in a predictable way. Across a range of studies, mean test loss has been observed to decrease as a power law in both the number of network parameters ($L \propto N^{-\alpha_N}$) and the number of training samples ($L \propto D^{-\alpha_D}$) [1, 2, 3, 4, 5, 6, 7]. Although aggregate performance changes smoothly with scale, when particular capabilities are examined, larger models often have emergent abilities, i.e., qualitatively different performance than smaller models [8, 9]. Understanding and reconciling both facets of scaling – the predictable power law decrease in loss and the emergence of new capabilities at scale – is of both theoretical and practical interest [10]. Understanding how scaling changes what neural networks learn is entangled with core questions: what are deep neural networks doing internally, and will they will continue to improve with scale?

Recent studies of the internals of neural networks have found a variety of impressive algorithms learned by gradient descent [11, 12, 13, 14, 15]. As more work is put into understanding the structures learned by neural networks (the task of *mechanistic interpretability*), we may find more and more *circuits* [11, 16] in models, intelligible internal algorithms for accomplishing prediction in specific contexts. Can such analysis be scaled up to frontier models [17]? Two assumptions which, if true, would make us more optimistic about mechanistically understanding large models include (1) decomposability/modularity/sparsity [18, 19, 20, 21] – that large models are decomposable into parts, and only a small number of these parts are relevant to the model's behavior on any given sample and (2) universality [22, 11, 23, 24] – that similar structures recur across models of increasing size. Recently, Olsson et al. [25] found encouraging evidence for universality of "induction heads" across LLMs and found that these emerge in a discrete transition during training.

In this paper, we articulate the *Quantization Hypothesis*, a set of informal conjectures about the *decomposability* of networks into smaller parts, the *universality* of computations performed across model scales, the *discreteness* of what models learn, and about how properties of the data distribution

---

[*]ericjm@mit.edu

[2]Project code can be found at: `https://github.com/ejmichaud/quantization-model`.

37th Conference on Neural Information Processing Systems (NeurIPS 2023).

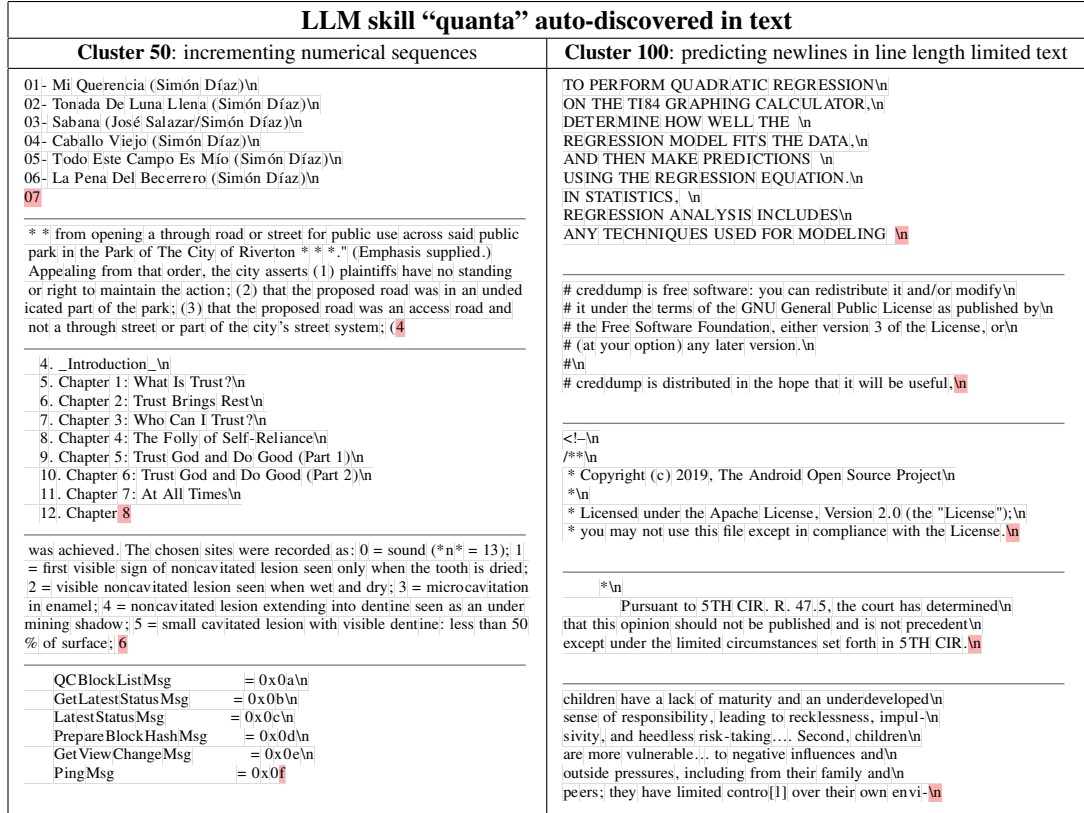

**LLM skill "quanta" auto-discovered in text**

| **Cluster 50**: incrementing numerical sequences | **Cluster 100**: predicting newlines in line length limited text |
|---|---|

```
01- Mi Querencia (Simón Díaz)\n
02- Tonada De Luna Llena (Simón Díaz)\n
03- Sabana (José Salazar/Simón Díaz)\n
04- Caballo Viejo (Simón Díaz)\n
05- Todo Este Campo Es Mío (Simón Díaz)\n
06- La Pena Del Becerrero (Simón Díaz)\n
07
```

```
* * from opening a through road or street for public use across said public
park in the Park of The City of Riverton * * *." (Emphasis supplied.)
Appealing from that order, the city asserts (1) plaintiffs have no standing
or right to maintain the action; (2) that the proposed road was in an unded
icated part of the park; (3) that the proposed road was an access road and
not a through street or part of the city's street system; (4
```

```
4. _Introduction_\n
5. Chapter 1: What Is Trust?\n
6. Chapter 2: Trust Brings Rest\n
7. Chapter 3: Who Can I Trust?\n
8. Chapter 4: The Folly of Self-Reliance\n
9. Chapter 5: Trust God and Do Good (Part 1)\n
10. Chapter 6: Trust God and Do Good (Part 2)\n
11. Chapter 7: At All Times\n
12. Chapter 8
```

```
was achieved. The chosen sites were recorded as: 0 = sound (*n* = 13); 1
= first visible sign of noncavitated lesion seen only when the tooth is dried;
2 = visible noncavitated lesion seen when wet and dry; 3 = microcavitation
in enamel; 4 = noncavitated lesion extending into dentine seen as an under
mining shadow; 5 = small cavitated lesion with visible dentine: less than 50
% of surface; 6
```

```
QCBlockListMsg          = 0x0a\n
GetLatestStatusMsg      = 0x0b\n
LatestStatusMsg         = 0x0c\n
PrepareBlockHashMsg     = 0x0d\n
GetViewChangeMsg        = 0x0e\n
PingMsg                 = 0x0f
```

```
TO PERFORM QUADRATIC REGRESSION\n
ON THE TI84 GRAPHING CALCULATOR,\n
DETERMINE HOW WELL THE  \n
REGRESSION MODEL FITS THE DATA,\n
AND THEN MAKE PREDICTIONS  \n
USING THE REGRESSION EQUATION.\n
IN STATISTICS,  \n
REGRESSION ANALYSIS INCLUDES\n
ANY TECHNIQUES USED FOR MODELING \n
```

```
# creddump is free software: you can redistribute it and/or modify\n
# it under the terms of the GNU General Public License as published by\n
# the Free Software Foundation, either version 3 of the License, or\n
# (at your option) any later version.\n
#\n
# creddump is distributed in the hope that it will be useful,\n
```

```
<!–\n
/**\n
 * Copyright (c) 2019, The Android Open Source Project\n
 *\n
 * Licensed under the Apache License, Version 2.0 (the "License");\n
 * you may not use this file except in compliance with the License.\n
```

```
    *\n
        Pursuant to 5TH CIR. R. 47.5, the court has determined\n
that this opinion should not be published and is not precedent\n
except under the limited circumstances set forth in 5TH CIR.\n
```

```
children have a lack of maturity and an underdeveloped\n
sense of responsibility, leading to recklessness, impul-\n
sivity, and heedless risk-taking…. Second, children\n
are more vulnerable… to negative influences and\n
outside pressures, including from their family and\n
peers; they have limited contro[l] over their own envi-\n
```

Figure 1: We auto-discover *quanta* – basic units of model knowledge/skill – for a language model. Here we show collections of next-token prediction samples which our method clustered together, each corresponding to some coherent model behavior. We indicate the token which was predicted from the context before it with a red highlight. We indicate newlines using "\n". See Section 5 for explanation.

produce power law neural scaling. In particular, we hypothesize that to many prediction problems, there corresponds a particular enumerable set of indivisible pieces of knowledge or skills that models must learn, and that model performance is determined by *which* of these elements models successfully learn. We call these basic building blocks of model performance the **quanta**:

> **Quantum (plural quanta)**: An indivisible computational module that, for example, retrieves a fact, implements an algorithm, or more generally corresponds to some basic skill possessed by a model.

We use this terminology in analogy to Max Planck's assumption in 1900 that energy is quantized into discrete chunks (quanta) – here we imagine that knowledge/skills are quantized into discrete chunks (quanta). Since "quantization" is commonly used in machine learning in the context of low-precision arithmetic, we suggest "knowledge quantization" or "skill quantization" to refer to our notion of quantization. We will see that a Zipfian distribution governing the "use frequency" of the quanta produces power law neural scaling, where the effect of scaling is to learn an increasing number of discrete quanta, and smooth scaling laws average over small discrete jumps in model performance.

This paper is organized as follows: in Section 2 we give a theoretical model of power law neural scaling from the Quantization Hypothesis. In Section 3 we construct toy datasets satisfying the hypothesis, where smooth power laws average over many discrete jumps in model performance. In Section 4 we then analyze how power law scaling decomposes for real LLMs. In Section 5, we develop a method for automatically discovering quanta in language models by clustering their behavior into basic coherent skills, and analyze the statistics of these clusters, concluding in Section 7.

## 2  Theory

Consider the task of modeling the distribution of text on the internet. Successful prediction requires an immense amount of knowledge, and the ability to perform diverse computations, due to the immense complexity and diversity of the world and therefore of human language. For instance, in order to predict what word will come next in a conversation between two physicists, one must "know" much about physics. In order to continue the text "2534 + 7261 = ", one must be able to perform arithmetic (for large enough numbers, memorization becomes a highly inefficient strategy) [26]. A great many distinct types of computations are present in the world in the processes that *produce* text, and so *predicting* text requires those computations to be present in our models.

In this paper, we conjecture the Quantization (or Quanta) Hypothesis:

> QH1 Many natural prediction problems decompose into an enumerable set of computations, pieces of knowledge, or skills, which models must learn to reduce loss. We call these **quanta**, and model them as being *discrete*, – they are either learned or not learned. Model performance is determined by *which* quanta have been learned.
>
> QH2 Some quanta are more useful for reducing loss than others, leading to a natural ordering of the quanta. We call the ordered quanta the **Q Sequence**. Optimally trained networks should therefore learn the quanta in that order. The effect of scaling is to learn *more* of the quanta in the Q Sequence, so scaling performance is simply determined by *how many* quanta are successfully learned.
>
> QH3 The frequencies at which the quanta are used for prediction follow a power law.

Together these can result in power law neural scaling. We model the Quantization Hypothesis as follows, referring to the below as the "Quantization (or Quanta) Model". Let $\mathbf{q}$ denote a bit string whose $k^{\text{th}}$ bit $\mathbf{q}_k = 1$ if the $k^{\text{th}}$ quantum in the Q Sequence has been learned, and $\mathbf{q}_k = 0$ otherwise. QH1 implies that the mean loss $L$ is simply a function of $\mathbf{q}$. QH2 implies that when $n \equiv \sum_k \mathbf{q}_k$ quanta have been learned, we have $\mathbf{q}_k = 1$ for $k \leq n$. Let $L_n$ denote the mean loss in this case. From QH3, we have that the $k^{\text{th}}$ quantum benefits prediction on a randomly chosen sample with probability

$$p_k = \frac{1}{\zeta(\alpha+1)} k^{-(\alpha+1)} \propto k^{-(\alpha+1)} \tag{1}$$

for a Zipf power law $\alpha > 0$, where $\zeta(s) \equiv \sum_{k=1}^{\infty} k^{-s}$. Let us also assume that learning the $k^{\text{th}}$ quantum reduces average loss from $b_k$ before it is learned to $a_k$ after it is learned on the samples where it is utilized. If $a_k$ and $b_k$ are $k$-independent ($a_k = a$, $b_k = b$), then a model that has learned the first $n$ quanta will have expected loss

$$
\begin{aligned}
L_n &= \sum_{k=1}^{n} a p_k + \sum_{k=n+1}^{\infty} b p_k = \sum_{k=1}^{\infty} a p_k + \sum_{k=n+1}^{\infty} (b-a) p_k \\
&\approx a + \frac{b-a}{\zeta(\alpha+1)} \int_n^{\infty} k^{-(\alpha+1)} dk = a + \frac{b-a}{\alpha \zeta(\alpha+1)} n^{-\alpha}.
\end{aligned}
\tag{2}
$$

In other words, $L_\infty = a$ and $(L_n - L_\infty) \propto n^{-\alpha}$ is a power law.

In Appendix A, we provide analogous derivations for other assumptions for $a_k$ and $b_k$, and find that a range of assumptions produce curves that are exact or approximate power laws – the latter include a small logarithmic correction.

In the derivation above, we assumed that all samples are what we will refer to as *monogenic*, meaning that prediction relies on at most a single quantum, akin to how monogenic traits in biology (e.g. cystic fibrosis) depend on a single gene. By assuming that all samples are monogenic, we can write the expected loss as a sum over quanta, weighted by the fraction of samples which rely on that quanta $p_k$. We further explore the idea of monogenic vs. polygenic samples in Section 4.2. So far we have seen how the Quantization Hypothesis can produce power law scaling as a function of the number of quanta learned $n$. We will now give one possible mechanism by which this can translate into power law scaling in parameters, data, etc.:

**Parameter scaling**: In networks of finite size, network capacity can bottleneck how many quanta are learned. If we assume that all quanta require the same capacity of $C$ network parameters, then a network with $N$ parameters can learn roughly $n \approx N/C$ quanta. Therefore $L(N) - L_\infty \propto n^{-\alpha} \approx (N/C)^{-\alpha} \propto N^{-\alpha}$, we so we get power law scaling in $N$ with exponent $\alpha_N = \alpha$.

**Data scaling (multi-epoch)**: For data scaling, we assume that for each quantum, a threshold of $\tau$ examples utilizing quantum $k$ are needed in the training set for quantum $k$ to be learned[3]. With $D$ training samples, approximately $Dp_k$ samples relying on quantum $k$ will be present, and solving for $Dp_n = \tau$ we get the last quantum to be learned will be $n \propto (D/\tau)^{1/(\alpha+1)}$ since $p_k \propto k^{-(\alpha+1)}$. Under this model, we get scaling in data samples $L(D) - L_\infty \propto n^{-\alpha} \propto (D/\tau)^{-\alpha/(\alpha+1)} \propto D^{-\alpha/(\alpha+1)}$, and so $\alpha_D = \alpha/(\alpha+1)$. From our earlier result that $\alpha_N = \alpha$, we would therefore predict that $\alpha_D = \alpha_N/(\alpha_N + 1)$. We discuss whether this relationship holds empirically for data and parameter scaling exponents observed across a variety of studies in Appendix F.

**Data scaling (single-epoch)**: In multi-epoch training, the information contained in the training dataset can bottleneck which quanta are learned. However, the rate of convergence of SGD can also bottleneck performance. For single-epoch training, a greater number of training samples allows one to train for longer. In our model, the amount that each quantum reduces mean loss by follows a power law. If the magnitude of the gradients for learning these quanta also follow a power law, then the convergence time for each quanta may follow a power law too. If the number of steps to learn quantum $k$ is $\propto 1/p_k$, then if the first quantum requires $T$ steps to be learned, quantum $n$ will require $Tn^{\alpha+1}$ steps, and so $n = (S/T)^{1/(\alpha+1)}$ quanta can be learned in $S$ steps. This gives scaling in training steps $L(S) - L_\infty \propto n^{-\alpha} \approx (S/T)^{-\alpha/(\alpha+1)} \propto S^{-\alpha/(\alpha+1)}$, and so $\alpha_S = \alpha/(\alpha+1)$. Under this model, multi-epoch and single-epoch data scaling exponents coincide: $\alpha_D = \alpha_S$.

## 3 Proof of concept: a toy dataset

In this section, we will describe a toy dataset consisting of distinct subtasks which are power law distributed in frequency. We observe power law neural scaling in data and parameters on this task, and find that the mechanism of neural scaling coincides with our theory from Section 2. It is therefore possible for scaling laws to arise from the Quantization Model for data with the right structure. We leave a study of whether natural datasets (e.g. natural modeling) possess such structure to Section 4.

### 3.1 The "multitask sparse parity" dataset

The toy task we will construct consists of many subtasks – distinct types of inputs which each require corresponding distinct computations (quanta). For each subtask, we choose a variant of the "sparse parity" problem, recently studied in [28]. The sparse parity prediction problem is simple: given a bit string of length $n$, compute the parity (sum mod 2) of a fixed subset of $k$ of those bits. We introduce an extension of this task, which we call "multitask sparse parity". Beyond $n$ and $k$, multitask sparse parity adds an additional parameter $n_{\text{tasks}}$, the number of subtasks (number of distinct versions of sparse parity) present in the dataset. To construct the task, we first choose $n_{\text{tasks}}$ random subsets $S_i$ of $k$ indices from $\{1, 2, \ldots, n\}$: $S_i \subset \{1, 2, \ldots, n\}$ and $|S_i| = k$, where $i = 1, 2, \ldots, n_{\text{tasks}}$. Input bit strings are length $n_{\text{tasks}} + n$. We call the first $n_{\text{tasks}}$ bits the *control bits* and the last $n$ bits the *task bits*. If control bit $i$ is active, then the parity is computed from the subset $S_i$ of the task bits. The control bits 1-hot encode the task number: on a given input, only one control bit is set to $1$ at a time – the rest are zero. For the sample shown below, since control bit $2$ is active, the answer is the parity of the task bits $S_2 = \{2, 7\}$, which is $0$ for this input:

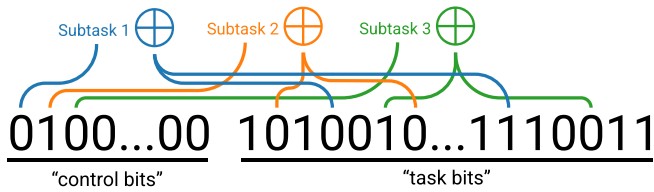

[3]This type of threshold has precedent, e.g. for the algorithmic tasks where "grokking" occurs [27].

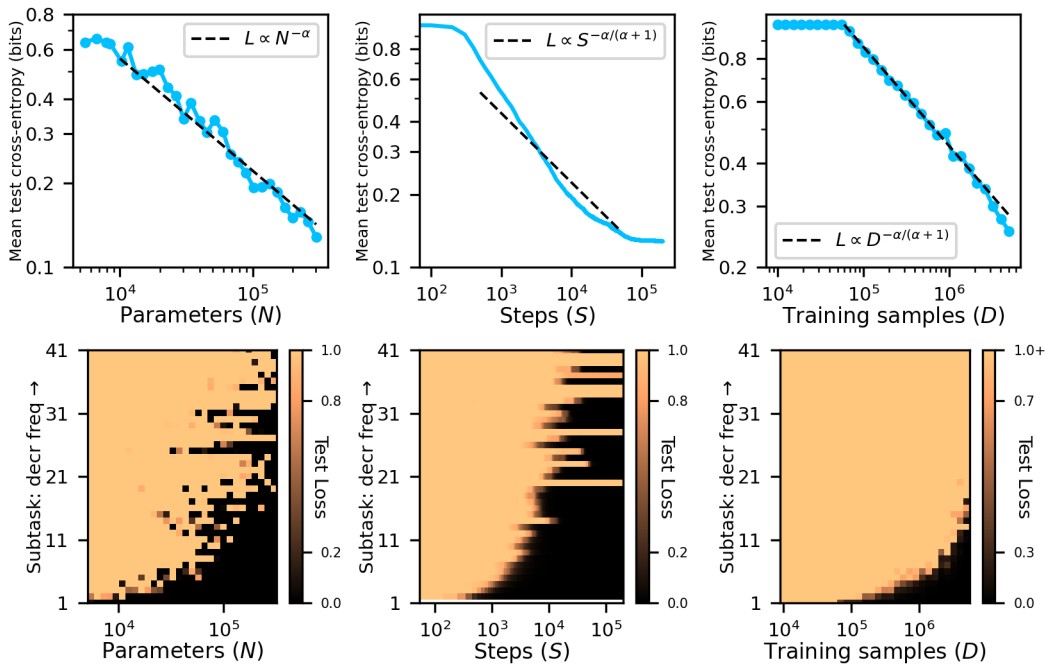

Figure 2: **Top:** Neural networks exhibit power law scaling in loss w.r.t. parameters $N$, training time $S$, and training samples $D$ (for multi-epoch training) when trained on the multitask sparse parity dataset. Here $\alpha = 0.4$ and we plot lines $\propto N^{-\alpha}$, $\propto S^{-\alpha/(\alpha+1)}$, $\propto D^{-\alpha/(\alpha+1)}$. **Bottom:** neural scaling broken down by subtask. Scaling behavior on individual subtasks exhibits emergence, where subtasks are suddenly learned above a particular scale. Power law neural scaling of mean test loss averages over a large number of qualitative changes in network performance (when broken down by subtask), with loss being driven to zero on an increasing number of subtasks which are power law distributed in frequency, a realization of the mechanism of neural scaling discussed in Section 2.

We impose a uniform distribution over the task bits. On the control bits, we impose a Zipfian distribution: the probability that a sample has control bit $i$ active (and therefore the parity must be computed from the subset $S_i$ of the task bits) is $\frac{1}{Z}i^{-(\alpha+1)}$ where $Z = \sum_{i=1}^{n_{\text{tasks}}} i^{-(\alpha+1)}$. This imposes a power law distribution over subtasks in data. Since answers are parities, this task can be treated as a binary classification problem on the subset of bit strings $\{0,1\}^{n_{\text{tasks}}+n}$ where for each string all but one bit of the first $n_{\text{tasks}}$ bits are zero.

## 3.2 Power law scaling and emergence

We train ReLU MLPs with a single hidden layer to solve this task with cross-entropy loss. The input dimension is $n_{\text{tasks}} + n$. We use the Adam optimizer with a learning rate of $10^{-3}$. To study scaling with respect to the number of model parameters, we train networks of varying width by sampling batches online. Within an individual single-epoch training run, we can study scaling in steps $S$. To study scaling with respect to multi-epoch training dataset size $D$, we use a network of sufficient width for capacity to not be a bottleneck, and for varying $D$ we sample a training set of $D$ samples and train for multiple epochs, recording model performance when mean test loss is lowest (early-stopping).

Training dynamics on the multitask sparse parity problem are highly nontrivial – on each individual subtask, loss follows a reverse-S curve, dropping after an initial plateau. This transition happens at different times for different subtasks, so the overall loss decreases smoothly, averaging over these transitions. See Appendix B for more discussion of training dynamics.

Figure 2 shows scaling curves on the multitask sparse parity problem. For the results shown, we used $n_{\text{tasks}} = 500$, $n = 100$, $k = 3$, $\alpha = 0.4$, and a batch size of 20000. We vary training dataset size from 1e4 to 5e6 and vary hidden-layer width from 10 to 500 neurons. We train for 2e5 steps. In line with the theory from Section 2, we find that as we scale training data and parameters, networks learn more

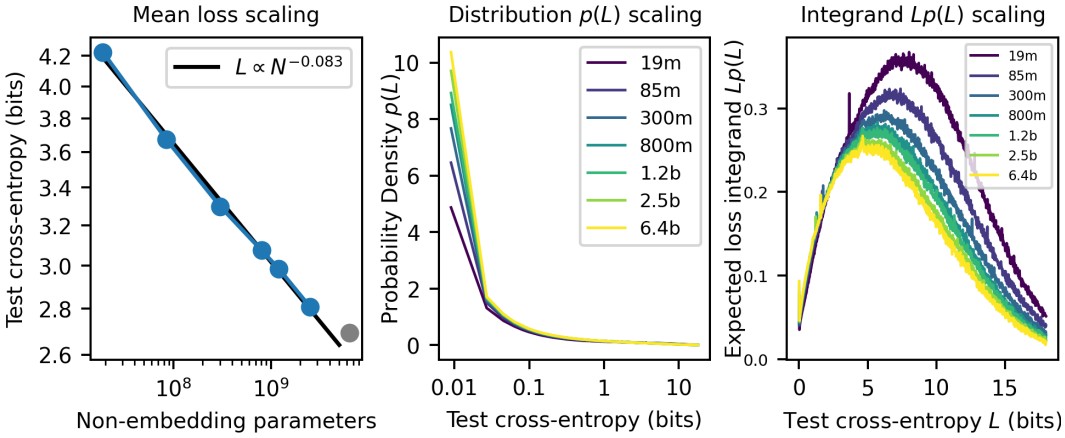

Figure 3: **Left:** Scaling of mean test loss w.r.t. non-embedding parameters for the Pythia models [29]. The parameter scaling exponent $\alpha_N$ is measured to be $\approx 0.083$ from the first six points along the curve (the seventh model appears to break the trend). **Center:** the distribution $p(L)$ over losses on individual samples for models of different size. Losses $\approx 0$ are by far the most common, and larger models achieve $\approx 0$ loss on an increasing fraction of samples. **Right:** the expected loss integrand $Lp(L)$ for models of different sizes. Low-loss samples contribute minimal mass to the mean loss, which is instead dominated by samples with much higher loss of 5-10 bits (depending on scale).

and more quanta (reducing loss on more and more subtasks), roughly in order of their frequency, and that this is what drives neural scaling. We see that scaling w.r.t. parameters is noisier than data scaling, possibly due to model initialization having some influence on which quanta are learned (for our data scaling experiments, we use the same seed and same model size for all runs). We also see that for scaling on individual subtasks, there is a rough scale of data or parameters below which networks do not learn the task, and above which they do. Smooth power law scaling therefore averages over a large number of emergent changes in model performance when properly decomposed by subtask, a proof of concept that the Quantization Model can be the mechanism of neural scaling for data with the right structure. See Appendix B for additional results and discussion on how the scaling exponents $\alpha_N, \alpha_S, \alpha_D$ relate to the subtask distribution power law exponent $\alpha + 1$ empirically.

## 4 Decomposing LLM scaling laws

We now study how scaling curves for large language models decompose. For our experiments, we use the Pythia model suite from Eleuther AI [29], a set of decoder-only transformers of varying size trained on approximately 300 billion tokens of The Pile [30]. We evaluate several models in the suite (ranging from 19m to 6.4b non-embedding parameters) on approximately 10 million tokens from the test set of The Pile. We record cross-entropy loss on every token, enabling us to study how loss on individual tokens, as well as how the distribution over losses, changes with model scale.

### 4.1 The distribution over per-token losses

In Figure 3, we show how the distribution over losses scales with model size. First, we find that for the first six models in the Pythia sequence, the mean loss scales as a power law with exponent $\alpha_N = 0.083$, roughly in line with the parameter scaling exponent of 0.076 measured in [3]. The 6.4b model does not fit the scaling curve well, so we excluded its loss when measuring the scaling exponent. Next, we plot the probability distribution over per-token losses $p(L)$. We find that losses close to zero are by far the most common, and that scaling increases the portion of approximately-zero losses. We also plot $Lp(L)$, the probability density over losses weighted by loss. The mean loss is the area under this curve. We see that despite approximately-zero-loss tokens being by far the most common, they do not contribute much mass to the mean loss. See Figure 11 for how these distributions change over training steps rather than across model size. We note that neural scaling in the wild is much more complicated than for multitask sparse parity – notably, the distribution over

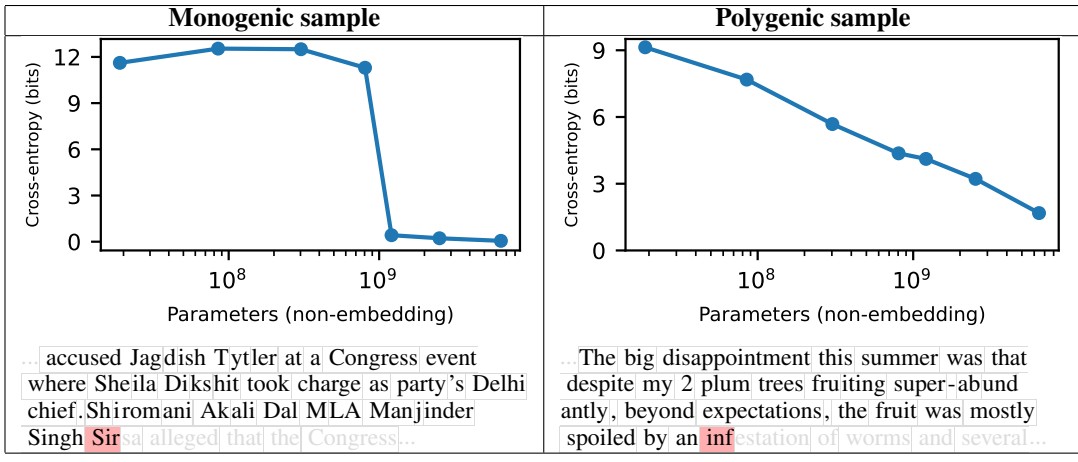

Figure 4: Per-sample scaling curves can have diverse behavior. Here we show extreme examples where scaling (of loss on predicting the highlighted token) is abrupt versus smooth. If the Quantization Hypothesis describes language modeling, then samples with sharp scaling would be *monogenic*, displaying sharp emergence at a particular model scale when the relevant quantum is learned. Samples with gradual scaling would be *polygenic*, where many quanta, emerging at different scales, marginally improve the loss. We show additional examples in Figure 12.

losses is not bimodal. We leave a detailed study of whether the statistics of neural scaling in LLMs are compatible with prior models of neural scaling to future work.

## 4.2   Monogenic versus polygenic scaling curves

In our introduction of the Quantization Hypothesis in Section 2 and our multitask sparse parity study in Section 3 we modeled network performance on individual samples as benefitting from a single quantum – all samples belong to a single subtask, which is either solved or not solved in a binary fashion. In our model and on multitask sparse parity, scaling curves on individual examples all exhibit emergence – loss on individual examples undergoes a sharp transition at a particular scale of parameters or data. Do we observe this in large language models?

Inspecting a large number of per-token (per-sample) scaling curves, we observe a variety of scaling behaviors. On some samples, loss drops at a particular scale. More typically though, loss improves at multiple scales. If the Quantization Hypothesis is true and the effect of scaling is to simply add new quanta to the model, then for per-sample loss curves to show progress at multiple scales, those samples must benefit from multiple quanta additively. As first mentioned in Section 2, we borrow terminology from genetics and refer to prediction problems for which the model's performance is determined by a single quantum as *monogenic* (akin to when a single gene determines a trait) and as *polygenic* when multiple quanta influence performance (in analogy to when multiple genes contribute to a trait). In multitask sparse parity, all prediction problems are monogenic. In natural language, we observe that model performance on most tokens improves at multiple scales, suggesting that most tokens are polygenic, but we can find tokens for which loss drops as a single phase transition in scale. Polygenicity forms a spectrum: the smoothness of the loss curve can vary substantially between examples, presumably with some prediction problems using few quanta and others using many. In Figure 4, we show extreme examples of both monogenic and polygenic samples.

Note that our monogenic/polygenic taxonomy of model behaviors assumes that QH1 and QH2 are true. However, it could be the case that there isn't an underlying discreteness to what is learned, or that scaling totally changes what networks learn, rather than simply adding additional quanta. Whether scaling truly has the effect we described will have to be investigated in future studies of the internals of neural networks. We also note that it is possible that sharp transitions in the per-token loss curves could be due to noise – if we had multiple runs with different random seeds for each model scale, we could better test whether the mean loss across seeds decreases smoothly or if there is a genuine discreteness where gradual progress is impossible for apparently "monogenic" tokens.

# 5 The quanta of language modeling

We have conjectured that the internals and behavior of language models are decomposable into an enumerable set of modules and associated skills (quanta). What might these basic building blocks of LLMs be? In this section, we develop a preliminary method to discover quanta. In particular, we will attempt to cluster tokens in a language corpus according to what knowledge or skill LLMs use to predict those tokens from their context. Our goal is to find coherent clusters of language model behavior that each reveal some distinct skill that the model has learned. Note that in *clustering* tokens to discover quanta, we are making the likely unrealistic assumption that these tokens are monogenic – that there is only one quantum involved in predicting each token. Not also that these clusters of behavior will not give us a mechanistic understanding of the quanta, but simply provide examples of LLM skills which could be studied further in future work.

We propose the use of gradients to cluster next-token prediction samples, where a "sample" consists of a token and its context in some document. Given some model, we will cluster two samples together if the gradient of the model's loss on each sample w.r.t. the model's parameters is similar for the two samples. The intuition for using gradients is as follows: if a model uses the same internal module to generate its prediction on two samples, then the gradients for parameters within the module may be nonzero and similar for the two samples (and possibly $\approx 0$ for parameters in irrelevant modules). If a model uses different modules to generate its prediction on different samples, then the gradients may not overlap. We therefore use gradient similarity as a proxy for *mechanistic similarity* – whether a model uses similar mechanisms/modules to generate its prediction on distinct samples. While crude, we find that gradients contain enough information to allow us to automatically discover many coherent clusters of LLM behavior using the following algorithm:

**Quanta Discovery from Gradients (QDG)**: We will use spectral clustering on gradients to find clusters of samples whose gradient has nonzero cosine similarity. Given a set of samples $(x_i, y_i)$ and a model $f_\theta$, we compute gradients for each sample $g_i = \nabla_\theta L(f_\theta(x_i), y_i)$. We then normalize these gradients $g_i \mapsto \hat{g}_i$ so that $\hat{g}_i \cdot \hat{g}_i = 1$. Let $A$ be a matrix whose rows are the normalized gradients: $A_{i,\cdot} = \hat{g}_i$. If we are clustering $d$ samples and our model has $n$ parameters, $A$ has shape $(d, n)$. We compute an affinity matrix $C = AA^T$, a matrix of shape $(d, d)$ where $C_{ij} = \hat{g}_i \cdot \hat{g}_j$, the cosine similarity between gradients $g_i, g_j$. From this, we compute an affinity matrix of the angular similarities $\hat{C}$ (which take values in $[0, 1]$) via $\hat{C}_{ij} = 1 - \arccos(C_{ij})/\pi$. We then perform spectral clustering with $\hat{C}$ to cluster samples.

QDG is expensive to compute for large models and for large numbers of samples. We therefore only apply it to the smallest model in the Pythia suite, which has 19m non-embedding parameters. We cluster 10000 tokens on which this model is confident and correct in its prediction, achieving less than 0.1 nats of cross-entropy. See Appendix C.1 for more detail.

We find that many, though not all, QDG clusters reveal some coherent model behavior. We show examples from clusters in Figure 1 and Figure 13. These clusters were found with the spectral clustering hyperparameter `n_clusters = 400`. While most clusters involve the prediction of the same token, manually inspecting these clusters we find that they usually involve predicting the same token for a coherent reason, rather than being based merely on having the same output. We also find clusters for more abstract prediction rules. For instance, the quantum shown in the left column of Figure 1 is the skill of incrementing a numerical sequence, and the examples involve predicting a variety of different tokens representing numbers.

## 5.1 The natural distribution over language modeling quanta

In our model, some quanta are more frequently used than others. If these frequencies follow a power law in accordance with the Quantization Hypothesis, then we may expect QDG cluster sizes to be governed by a power law. The measured scaling exponent of $\alpha_N = 0.083$ from Figure 3 implies a power law distribution over quanta with exponent $-1.083$. Do the cluster sizes follow this?

Figure 5 shows rank-frequency curves for clusters discovered with QDG for varying choices of `n_clusters`. These curves sort the clusters according to their size and then plot size against cluster index (rank). We plot rank-frequency curves for many choices of `n_clusters` since it is unclear a priori which `n_clusters` to use. When we measure the slope of the rank-frequency curve, we measure it from the envelope formed by the many rank-frequency curves, a practice which we

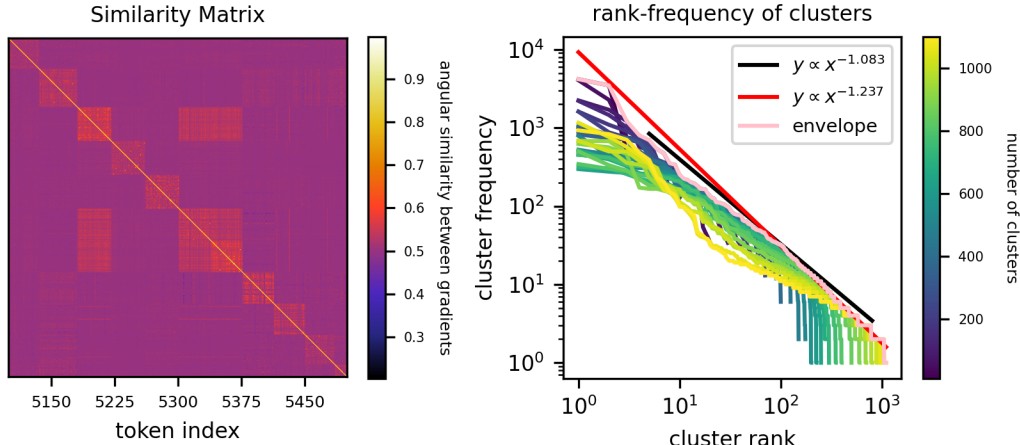

Figure 5: **Left:** angular similarity between model gradients for a variety of natural language samples. Samples are reordered according to their QDG cluster (with 400 clusters) to reveal the block-diagonal structure of the similarity matrix. We visualize a small part of the overall similarity matrix in this plot – note that not all clusters are as visibly distinct as the ones shown. **Right:** rank-frequency plot of QDG clusters. We measure the slope of the envelope of the rank-frequency curves from cluster rank 100-1000 to be $\approx -1.24$, which is a steeper than the slope of -1.08 expected from the measured parameter-scaling exponent from Figure 3, though within the margin of error given the uncertainty of our clustering methodology. See Appendix E for a discussion of the bias/uncertainty of our method.

discuss in Appendix E. Biases in the clustering algorithm and inherent noise in model gradients make clustering imperfect, and lead to high uncertainty of our the measured power law exponent. From our analysis in Appendix E, we think that extracting the power law exponent over quanta utilization frequency by measuring the slope of the rank-frequency curve should have uncertainty of at least 0.2. We also note that some rank-frequency curves don't look like a clean power law. In Appendix E, Figure 16 we find that we can get similar-looking curves in a toy model of this clustering process when the dimension and noise is high. Between ranks 100-1000, we measure a slope of $\approx -1.24$, about 0.16 off our expected slope of $-1.08$, and so within the margin of error. We are encouraged that the size of our discovered clusters seem to decay at a rate (very roughly) compatible with observed neural scaling exponents, in line with our theory. However, less naive clustering schemes, operating on more samples with more clusters, could be useful to sharpen this measurement.

## 6 Related Work

**Models of neural scaling**: Several models of neural scaling laws have been proposed in prior work. Sharma and Kaplan [31] explain power law scaling w.r.t. model parameters using an argument from approximation theory, which relates neural scaling exponents to the dimension of the data manifold $d$. Michaud et al. [32] point out that effective dimension $d$ could be generalized to the maximum arity of the target function's computation graph for sparse compositional problems. Bahri et al. [33] generalized the model of Sharma and Kaplan [31] to scaling w.r.t. dataset size, additionally relating scaling exponents to the power law spectrum of certain kernels. Maloney et al. [34] develop an exactly solvable random-feature model of scaling, from which they derive a joint parameter-data scaling law. Bordelon et al. [35] develop a model of data scaling for kernels, decomposing the generalization error into a sum over eigenmodes, whereas we decompose error into a sum over quanta. Arguably the closest prior work to ours is Hutter [36], who develops a model of data scaling wherein a discrete set of "features" must be learned. In this model, a feature is learned if it occurs at least once in the training set. If the features are Zipfian distributed, this produces power law scaling in expectation but with high variance. In our model, using a data threshold $\tau \gg 1$ lowers the variance in the scaling curve, and we also considered scaling w.r.t. parameters and applied the model to real networks.

**Understanding emergent abilities**: Wei et al. [8] and Srivastava et al. [37] document examples of emergent abilities in large language models, though Schaeffer et al. [38] suggest that these examples

are an artifact of the metric used to evaluate performance. Arora and Goyal [39] develop a framework for the emergence of "skills", where predicting text requires combining multiple different skills from an underlying set of language skills.

**Miscellaneous**: The topic of phase transitions in machine learning is not new [40], but our work was strongly influenced by the recent work of Olsson et al. [25] who observe a phase change from the formation of induction heads and especially Nanda et al. [13] who conjecture that phase changes may be ubiquitous. Simon et al. [41] also exhibit a task where learning proceeds as a series of discrete steps. Chen et al. [42] develop a framework for understanding LLM "skills" in a hierarchy and for choosing data to more efficiently learn desired skills. Chan et al. [43] study how a Zipfian data distribution influences in-context learning.

# 7   Discussion

**Summary**: The Quantization Hypothesis posits that for some types of prediction problems, models must learn a discrete (quantized) set of modules/knowledge/skills (quanta). When data is distributed in such a way that the "use frequencies" of these quanta follow a power law, then power law neural scaling can arise as models learn more and more quanta, with smooth scaling curves averaging over many small cases of emergence. We presented a toy dataset where neural scaling exhibits these properties. We then documented how language model scaling curves decompose, beyond simply how the mean loss scales. Lastly, we developed a method to discover quanta from the internal structure of trained models, from which we were able to enumerate a large number of skills of a small language model. The frequencies at which the quanta we discover are used for prediction in natural text seem to roughly track the power law our theory would predict, though this measurement is quite imprecise.

**Limitations**: While the Quantization Hypothesis appears to hold for our toy datasets, much work remains in investigating to what extent it holds for natural tasks like language modeling. Probably our riskiest assumption was that there is an underlying discreteness to *everything* that models learn. Gradual scaling seems typical in LLMs [38], and it could be more parsimonious to model neural scaling as an underlying smooth process rather than to assume that most tasks are highly polygenic with underlying discrete quanta. Note also that in our model of scaling w.r.t. parameters $N$, having more parameters merely increases the capacity of the network. In practice however, larger networks are more efficient learners [7], and one can trade off between parameters and data, whereas in our model parameters and data independently bottleneck the number of quanta that can be learned. Additionally, we modeled the quanta as being independent, where learning order is given just by the use frequencies, but it could make more sense to think of the quanta as living in a hierarchical dependency graph. Lastly, our QDG method is neither very principled nor scalable, and much better methods could likely be developed to discover quanta and study their statistics for larger models and across more samples.

**Implications for emergence and forecasting**: Srivastava et al. [37] find that on some tasks, neural scaling has high "linearity", with gradual improvements to scale, with other tasks displaying "breakthroughness", where performance improves sharply at some scale. In our model, high linearly would result from a task's relevant quanta being widely spread along the Q Sequence, and high breakthroughness would result from a task being monogenic or from the relevant quanta being close together in the Q Sequence. Our model also suggests that future capabilities could be forecasted if one could estimate the frequency at which that skill would benefit prediction in the training corpus.

**Implications for mechanistic interpretability**: If the Quantization Hypothesis is correct, then understanding a network reduces to enumerating its quanta. Having done this, the quanta could perhaps then be translated into a more interpretable format (something like code), studied in this format, and eventually executed in this format, rather than via the operation of the network.

**Outlook**: Lastly, our decomposition of networks into quanta is reminiscent of Minsky's *Society of Mind* [44] perspective that minds are decomposable into individually mindless "agents". If this decomposition is indeed possible, then the quanta (agents) become natural objects of study within networks. This *mesoscale* understanding of networks, in terms of the internal modules which collectively constitute their performance, could perhaps act like statistical physics for deep learning, allowing us to bridge our microscale understanding of low-level training dynamics and our macroscale understanding of model performance.

## Acknowledgments and Disclosure of Funding

We thank Tamay Besiroglu, Neel Nanda, Tony Wang, David Bau, Ben Edelman, Brian Cheung, Wes Gurnee, Stephen Casper, Peter Hase, Davis Brown, Eleni Shor, Max Nadeau, and Xander Davies for helpful conversations and feedback. We thank Lauro Langosco for helping with code to visualize samples from The Pile. This work was supported by the Foundational Questions Institute, the Rothberg Family Fund for Cognitive Science, the NSF Graduate Research Fellowship (Grant No. 2141064), and IAIFI through NSF grant PHY-2019786.

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

# Appendix

## A More general scaling laws

If one learns the first $n$ quanta, reducing the loss from $b_k$ to $a_k$ ($1 \le k \le n$), while the loss remains $b_k$ for $k > n$, the expected loss is given by:

$$L_n = \sum_{k=1}^{n} a_k p_k + \sum_{k=n+1}^{\infty} b_k p_k. \tag{3}$$

In the main text, we used $a_k = a$ and $b_k = b$ for our model. However, one can imagine a variety of other choices for $a_k$ and $b_k$.

**Case 1** $b_k = -\log p_k$ and $a_k = 0$, where $p_k = k^{-(\alpha+1)}/\zeta(\alpha+1)$. This baseline for $b_k$ is the error of a model which outputs the token frequencies, independent of the context (assuming that quanta involve the prediction of a particular token). The expected loss is given by:

$$L_n = \sum_{k=1}^{n} 0 \cdot p_k + \sum_{k=n+1}^{\infty} (-\log p_k) \cdot p_k \approx \frac{1 + \alpha + \alpha \log \zeta(\alpha+1)}{\alpha^2 \zeta(\alpha+1)} n^{-\alpha} + \frac{\alpha+1}{\alpha \zeta(\alpha+1)} n^{-\alpha} \log n, \tag{4}$$

which contains a power law term $n^{-\alpha}$ plus a log term $n^{-\alpha}\log n$. For very large $n$, the log term can be ignored, so $L$ is still approximately a power law of $n$ with exponent $-\alpha$, shown in Figure 6.

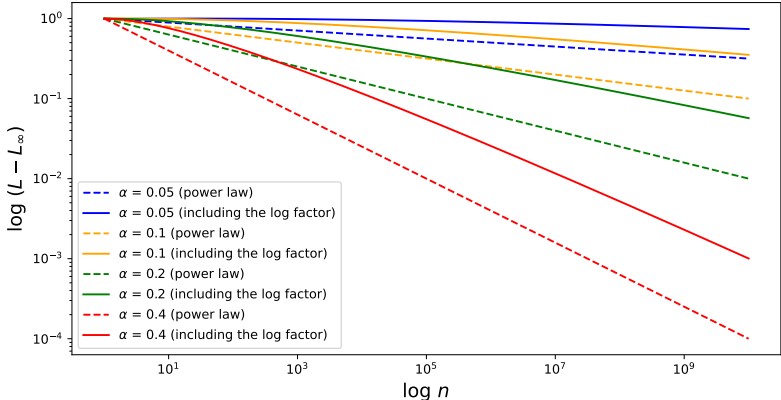

Figure 6: Comparing different scaling laws. Setting $a_k = 0$, we compare $b_k = -\log p_k$ (solid lines) and $b_k = 1$ (dashed lines) for different alphas. Although the $b_k = -\log p_k$ case would cause an extra loss term $n^{-\alpha}\log n$ in additional to the power law term $n^{-\alpha}$, the loss becomes a power law asymptotically when $n$ becomes large.

**Case 2** $b_k = -\log p_k$ and $a_k = -\log(Cp_k)$ ($C > 1$), where $p_k = k^{-(\alpha+1)}/\zeta(\alpha+1)$. The expected loss is given by:

$$L_n = \sum_{k=1}^{n} (-\log(Cp_k)) \cdot p_k + \sum_{k=n+1}^{\infty} (-\log p_k) \cdot p_k \approx \frac{\log C}{\alpha \zeta(\alpha+1)} n^{-\alpha} - \log C + \frac{1 + \alpha + \alpha \log \zeta(\alpha+1)}{\alpha^2 \zeta(\alpha+1)}, \tag{5}$$

which is a power law $n^{-\alpha}$ plus a constant.

## B Additional results on multitask sparse parity

**Training dynamics**: When loss is broken down by subtask on multitask sparse parity, learning curves consist of many reverse-S shaped curves, and mean loss decreases smoothly as an average over these curves. In Figure 7, we show loss versus time for each subtask for training runs in both the single-epoch and multi-epoch regimes. In Figure 8 we show how convergence time for each subtask relates to the frequency of that subtask.

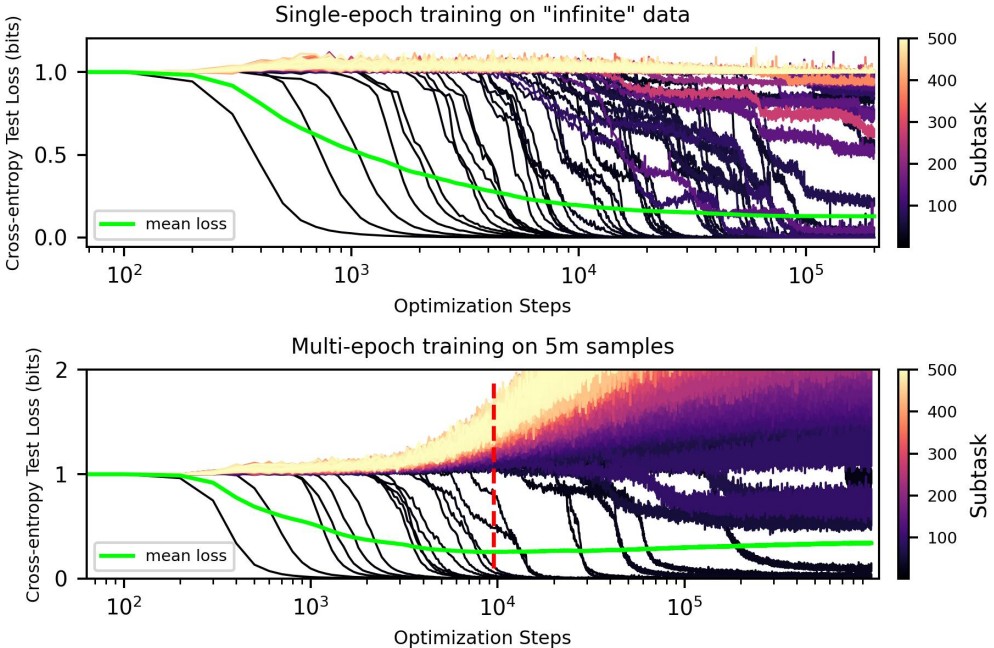

Figure 7: Training dynamics on the multitask sparse parity dataset consist of many "phase transitions" when decomposed by subtask – the loss curve for each subtask drops following an initial plateau of no apparent progress, in line with [28]. The mean loss decreases smoothly, averaging over these phase transitions in the model's performance on subtasks. We show curves for single-epoch training (top) and multi-epoch training on 5 million samples (bottom). The dashed red line indicates the early stopping point where mean test loss is minimized. For these runs, $\alpha = 0.4$.

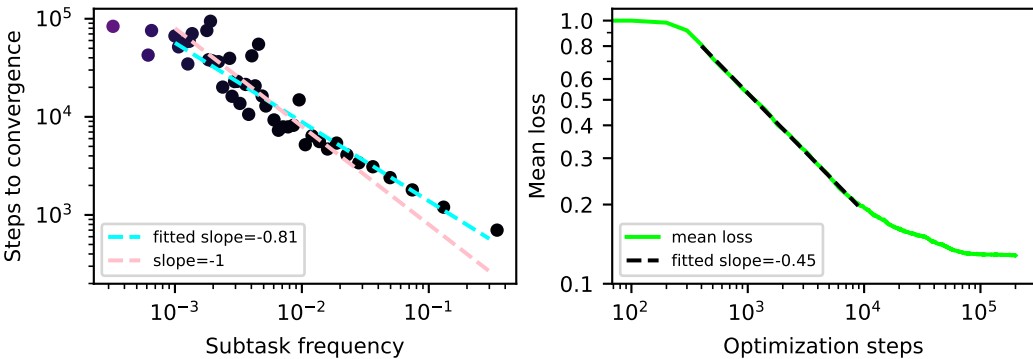

Figure 8: Convergence time for each subtask versus the frequency of that subtask. We see that convergence time $S_k$ on subtask $k$ is $S_k \propto p_k^{-0.81}$ rather than $S_k \propto p_k^{-1}$ as we had expected. This leads to a steeper scaling w.r.t. $S$ than expected from theory. For these experiments, we used $\alpha = 0.4$, and so we would have predicted $\alpha_S \approx 0.29$ but instead we get $\alpha_S \approx 0.45$. We consider the model to have converged on a subtask once it gets mean test loss less than 0.1 bits on that subtask.

**Scaling for varying** $\alpha$: In Figure 10 we show scaling curves on multitask sparse parity in $N, S, D$ for a variety of quanta distribution parameters $\alpha$. While all scaling curves appear to be power laws, the relationship between $\alpha_N, \alpha_S, \alpha_D$ and $\alpha$ is not precisely as predicted by theory:

1. **Parameter scaling:** We observe that the relationship between $\alpha_N$ and $\alpha$ deviates a bit from the prediction $\alpha_N = \alpha$, with $\alpha_N < \alpha$ for small $\alpha$ and $\alpha_N > \alpha$ for large $\alpha$. Perhaps model size does not influence learning just by changing capacity, but also by affecting optimization.

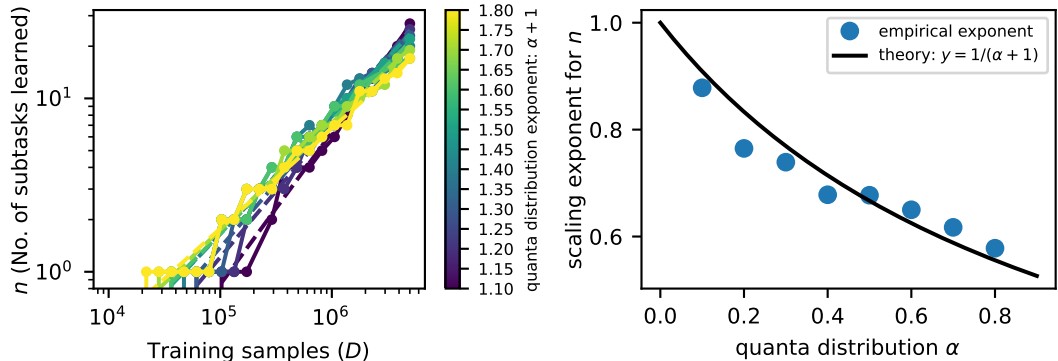

Figure 9: Number of subtasks learned ($n$), including subtasks learned after early-stopping would have terminated the training run, versus training samples $D$ for a variety of $\alpha$. We see that the relation $n \propto D^{1/(\alpha+1)}$ approximately holds, in line with theory. Deviation from theory for the scaling exponent of loss $L$ w.r.t. $D$ therefore likely originates from our failure to regularize network training, leading to early-stopping ending training before some subtasks can be learned.

2. **Step scaling:** We observe that $\alpha_S$ is consistently higher than the theoretical prediction $\alpha/(\alpha+1)$. In Figure 8, we saw that the number of steps to convergence for each subtask did not precisely follow $S_k \propto p_k^{-1}$, but was closer to $S_k \propto p_k^{-0.81}$. This means that many subtasks converge faster than we would expect, producing a steeper scaling curve.

3. **Data scalaing:** We observe that $\alpha_D$ is substantially higher than the theoretical prediction $\alpha/(\alpha+1)$ for small $\alpha$. We think this may be related to the fact that early-stopping cuts off training before all subtasks are learned as observed in Figure 7. In Figure 9, we show how the number of subtasks learned $n$, when we include subtasks learned after early-stopping, seems to be in line with theory: $n \propto D^{1/(\alpha+1)}$.

Better understanding the precise nature of power law scaling on multitask sparse parity is an interesting avenue for future work.

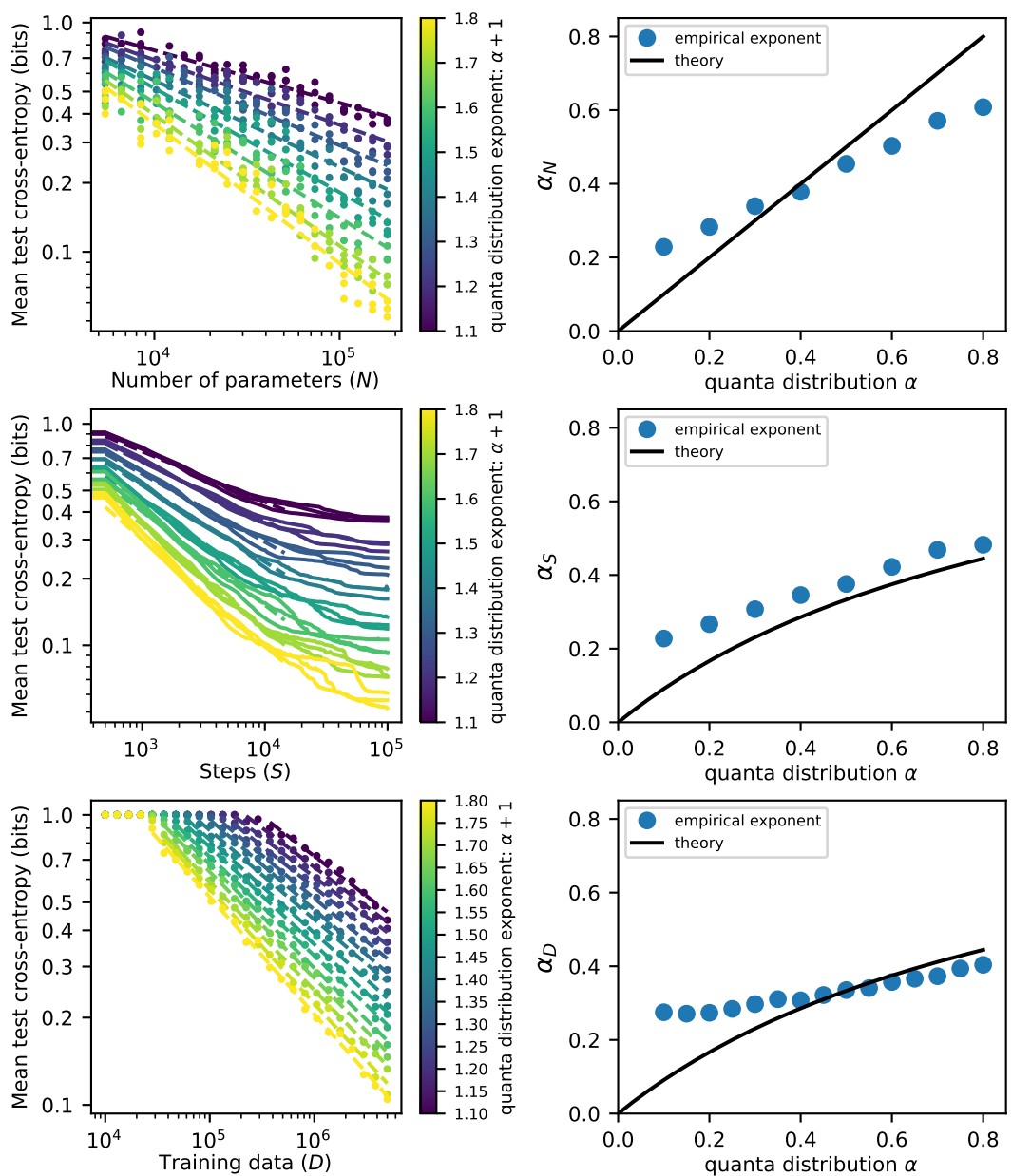

Figure 10: Scaling in parameters ($N$), single-epoch training time ($S$), and multi-epoch training samples ($D$) for varying quanta power law distribution parameter $\alpha$ on multitask sparse parity. We notice that scaling curves in steps $S$ are typically steeper than the $\alpha_S = \alpha/(\alpha + 1)$ predicted from theory, and that for low $\alpha$ the scaling curves in $D$ also deviate from theory substantially.

## C Additional results on language models

In Figure 11 we show how the distribution over losses changes across time during a training run, rather than across model scales like in Figure 3.

In Figure 13 we show additional examples from clusters discovered with QDG.

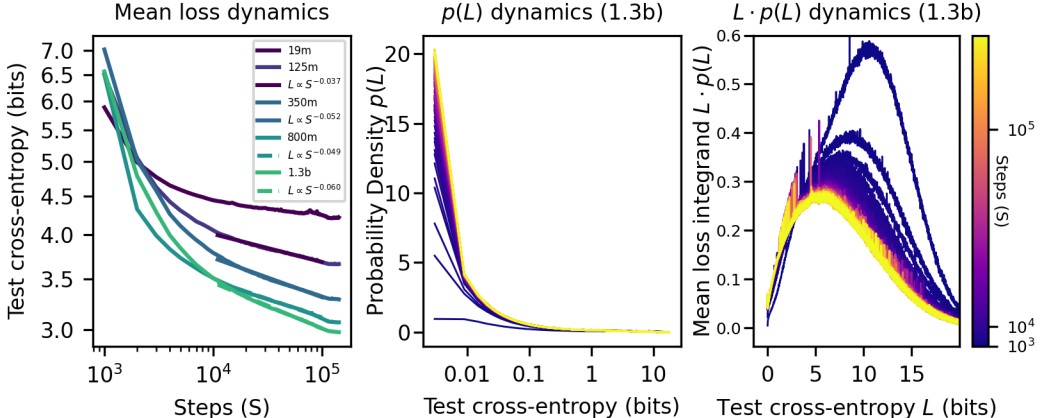

Figure 11: **Left:** Training curves (scaling w.r.t. steps $S$) of mean test loss for Pythia models. We measure exponents $\alpha_S$ between 0.037 and 0.06. **Center:** the distribution $p(L)$ over time. Over time, models achieve $\approx 0$ loss on an increasing fraction of tokens, similar to scaling in model size. **Right:** The distribution $L \cdot p(L)$ over time.

### C.1 Details of application of QDG to LLMs

When applying QDG to language models, we use gradients within self-attention and MLP layers, but do not include embed, unembed, or layer norm gradients when we flatten and concatenate gradients into a vector $g$.[4] We choose samples $(x_i, y_i)$ for which our 19m-parameter model achieves a cross-entropy loss less than 0.1 nats. We filter based on this criteria since (1) we cannot cluster samples based on model mechanism if the model does not have such a mechanism for performing prediction correctly on those samples and (2) our intuition that samples with particularly low loss are more likely to be monogenic. We further exclude samples which can be solved via induction on the context[5], since such samples are quite common (possibly interfering with our task of finding diverse quanta) and since early experiments indicated that QDG had trouble clustering such samples together. We choose 10000 such samples to perform clustering on from the test set of The Pile. After computing the affinity matrix $\hat{C}$, we use the spectral clustering implementation from scikit-learn [45] with labels assigned via k-means.

## D Quanta discovery on TinyStories

We also apply QDG to TinyStories-33M, a language model trained on the TinyStories dataset [46]. We consider only tokens on which TinyStories-33M achieves a loss less than 1 bit of cross-entropy. We apply QDG to 10000 such samples, clustering their gradients with spectral clustering with `n_clusters = 400`. We show some samples from the resulting clusters in Figure 14. Many of these clusters reflect some simple recurring pattern in the TinyStories dataset, like predicting " time" after "Once upon a", which many documents in the dataset start with. Some other clusters are more interesting however, like Cluster 11, which seems to involve predicting the correct noun in a sentence where that noun was referred to earlier in the sentence or in previous sentences.

---

[4]We exclude gradients for embed and unembed parameters because they are high dimensional and also because they may contain information more about the input and output rather than the computations the model performs internally. We exclude layer norm gradients because they appeared to contain less information about clusters in toy experiments.

[5]We filter (copying) induction problems by excluding samples where the token which is to be predicted is the last token in a trigram which occurred earlier in the context. This is not a very comprehensive filtering scheme.

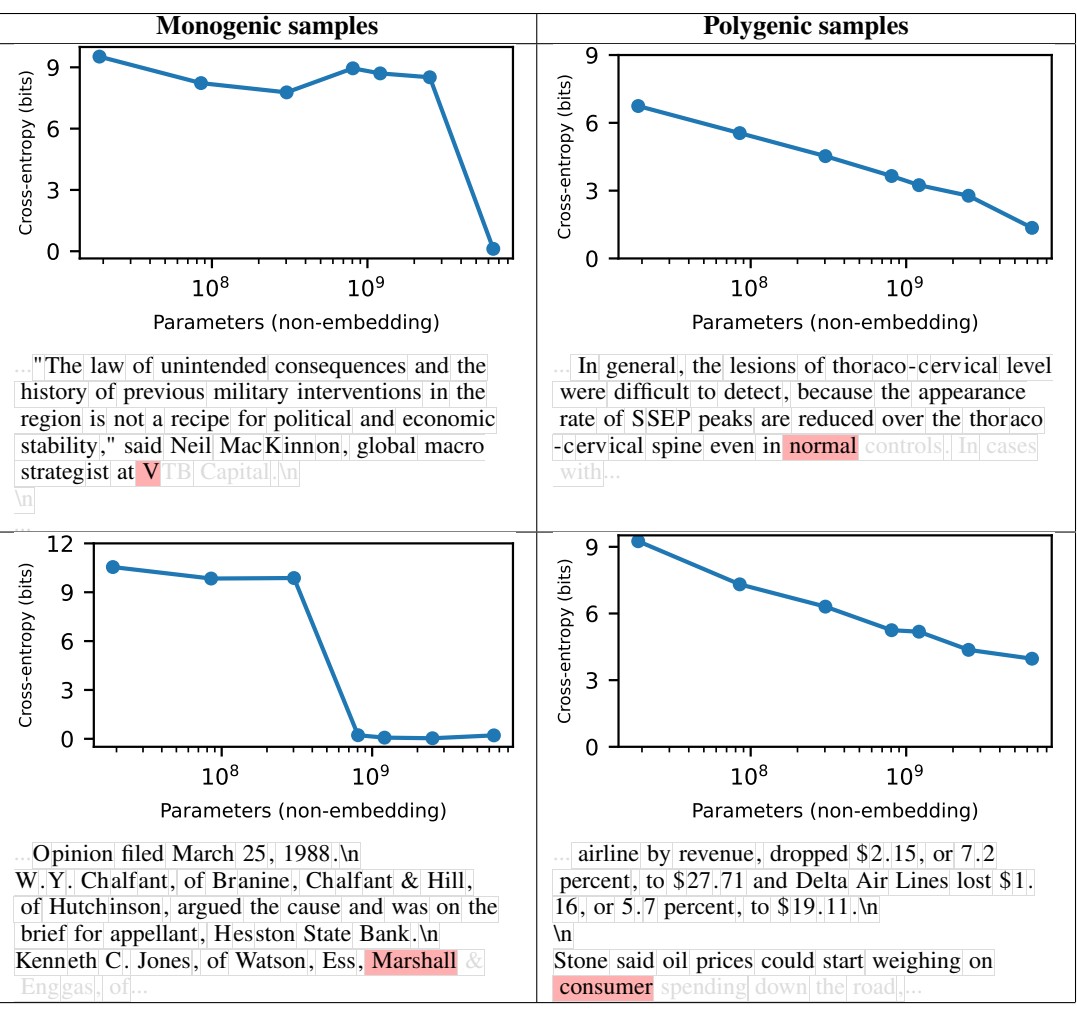

Figure 12: Additional LLM scaling curves on individual samples which exhibit sharp vs smooth improvement. If the Quantization Hypothesis is true for language modeling, then we would interpret samples with sharp drops as "monogenic" and samples with gradual progress as "polygenic".

# E   The difficulty of estimating the power law exponent from clusters

In Section 5.1, when we looked at the distribution over elements in each cluster, we did not perfectly recover a Zipf distribution with exponent $\approx 1.08$ that we expected from our theory. In this section, we describe the difficulty of accurately estimating such an exponent with our method.

## E.1   QDG on multitask sparse parity

As a first experiment, we performed QDG on multitask sparse parity, where there is a known, artificially-imposed power law distribution over subtasks. We train a width-500 single-hidden-layer ReLU MLP on multitask sparse parity with $\alpha = 0.4$ and with $n = 100$, $k = 3$, and $n_{tasks} = 500$. We then took 10000 samples which the network achieves $\approx 0$ loss on (sampled from the Zipf distribution over subtasks with exponent 1.4). We compute gradients of cross-entropy loss w.r.t. all model parameters for these samples, and then perform QDG just like for LLMs. We show results in Figure 15. We plot the full similarity matrix where samples are ordered according to their a priori known subtask, rather than their cluster from QDG, and see a clear pattern where elements from the same subtask have on average higher angular similarity than elements between subtasks. However, from the rank-frequency plot of the clusters, we do not recover a slope of -1.4, but rather a lower slope of $\approx -1.1$. This shows that even when there is an exact decomposition of inputs into subtasks with a

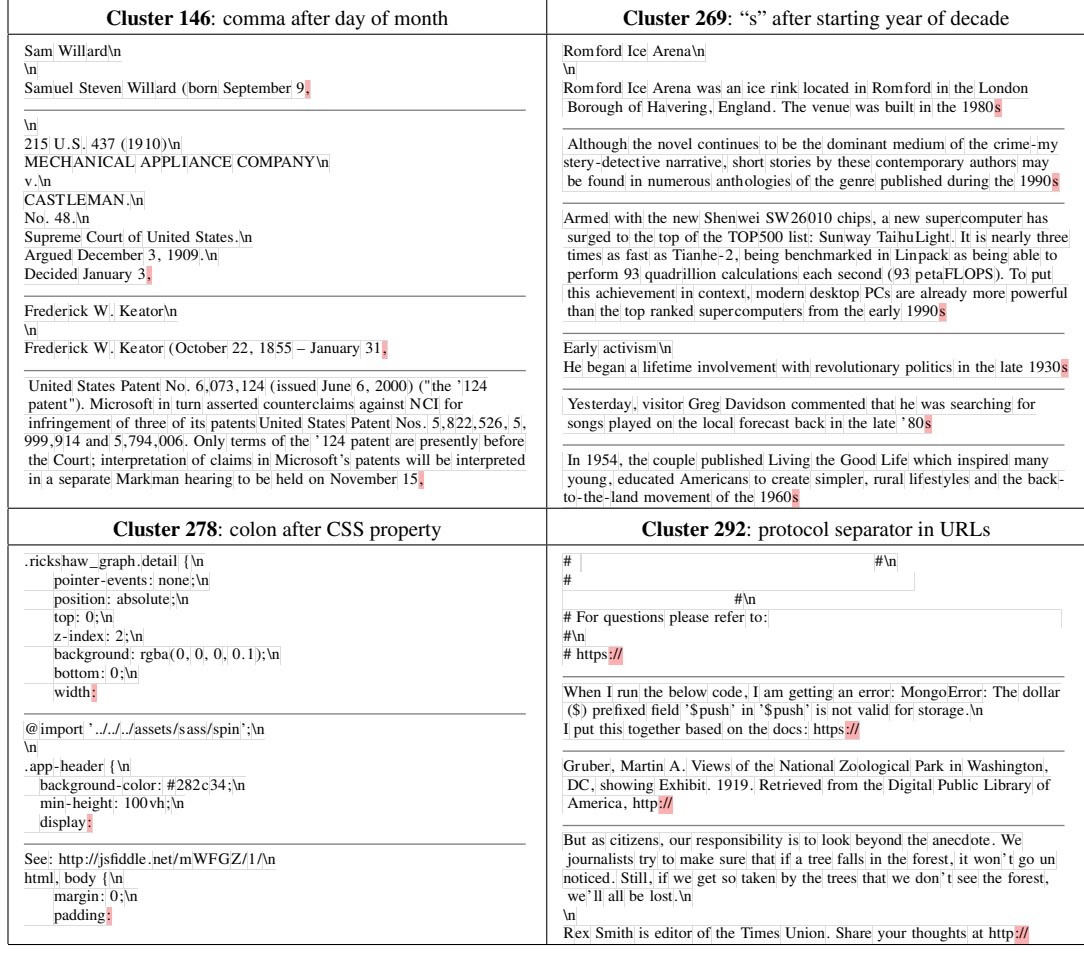

Figure 13: Additional examples of clusters of inputs discovered by QDG. Like in Figure 1, we used 10000 samples and `n_clusters` of 400.

known Zipf distribution over these subtasks, that we do not perfectly recover this Zipf distribution from QDG.

### E.2 A toy model of QDG uncertainty and bias

**A toy model:** To understand the bias of spectral clustering, we develop the following toy model. We assume the dataset has $N = 1000$ subtasks, each subtask containing $n_i = \lfloor \frac{A}{i^\alpha} \rfloor (1 \le i \le N)$ tokens ($A = 1000$). We use a Gaussian distribution $\mathcal{N}(\mathbf{m}_i, \sigma^2 \mathbf{I}_{d \times d})$ to model gradients within a subtask $i$, where $d$ is the embedding dimension, $\sigma$ is the noise level, and $\mathbf{m}_i$ is the Gaussian mean. $\mathbf{m}_i$ itself is drawn from the standard Gaussian distribution $\mathbf{m}_i \sim \mathcal{N}(\mathbf{0}, \mathbf{I}_{d \times d})$. We define the similarity between two vectors $\mathbf{x}, \mathbf{y}$ to be $\text{sim} \equiv 1 + \frac{\mathbf{x}}{|\mathbf{x}|} \cdot \frac{\mathbf{y}}{|\mathbf{y}|}$. We compute pairwise similarity between all $\sum_{i=1}^{N} n_i$ tokens, and input the similarity matrix to the spectral clustering algorithm. We also need to specify the number of clusters $k$.

We have two hyperparameters in the toy model, the embedding dimension $d$ and the noise level $\sigma$. We need to determine them such that this toy model can decently imitate LLM results (Figure 5). We fix $\alpha = 1$, sweeping $d = \{30, 100, 1000\}$, $\sigma = \{0, 0.5, 2.0\}$, and $k = \{100, 200, 500\}$. As shown in Figure 16, the high-dimension ($d = 1000$) large-noise ($\sigma = 2.0$) scheme seem to best agree with the LLM results, since the $k = 200$ curve can reproduce the sag and the cliff present in LLM curves.

Estimating $\alpha$ from the frequency curve is hard, in fact, the slope depends on $k$ and the region used to estimate it. However, we observe that different $k$ curves form a clear envelope, whose slope is

**Example "quanta" for the TinyStories dataset**

**Cluster 11**: predicting the correct noun

Lily asked her mom, "Can I touch the sunflower?" Her mom replied, "No, Lily. The sunflower is not for touching. It's for looking at." \n
\n
Lily was sad, but she understood. The next day, Lily rode her bike past the sun

Once upon a time, there was a big house with a door. The door was brown and it could move when people opened it. One day, a little boy came to the house and he saw the impressive door. He wanted to open it and see what was inside. So he moved the door

When they got to the park, Tim saw a man with a sack. The man had found Tim's wagon and put it in the sack to keep it safe. Tim was happy to have his wagon back and thanked the man. He put his wagon

Lily and Ben went to the park with Mom. They saw a big pond with many ducks and swans. Lily liked the swans. They were white and graceful. She wanted to feed them some bread.\n
\n
"Mom, can I give some bread to the sw

Lily and Ben nod. They promise to be careful. They ask mom to read the letter to them. Mom smiles. She reads the letter. It is from grandma. She says she loves them a lot. She sends them kisses and hugs. Lily and Ben are happy. They send kisses and hugs back to grandma. They thank mom for the letter

**Cluster 31**: " time" after "Once upon a"

Once upon a time

Once upon a time

Once upon a time

Once upon a time

Once upon a time

Once upon a time

Once upon a time

Once upon a time

**Cluster 75**: comma after temporal phrase

One day, Jack wanted to show off his cool sunglasses. He spotted a nice patch of grass in the park, and he decided to sit down and enjoy the sun. He put his sunglasses on and just sat. He felt so special.\n
\n
A few moments later,

Jack happily agreed and they started the game. At first, it was a bit tricky for both of them to get the squash to the other, but after a few tries, Jack was a pro. He laughed and cheered as he ran back and forth to get the squash. \n
\n
Mommy and Jack played until the sun started to go down. Then,

Bob loves to go on adventures. He went for a walk along the beach, looking for fun and exciting things. All of a sudden,

Linda was a little girl who loved wandering around in nature. She was eager to explore and find what she could. One day, Linda was wandering around in the woods when she spotted a big yellow flower. She was so excited that she ran over to it. She scooped the flower up and inspected it more closely. Soon,

Next, her mom told Emma to wipe the floor clean. Emma grabbed a cloth and wiped the floor. When she was finished, it was as clean as a new penny.\n
\n
Finally,

**Cluster 77**: beginning of quote

Jack showed her the big blue ticket and said, "This is my ticket. I'm going to lay it down."\n
The girl asked, "Where will you lay it down?"\n
Jack answered, "

When she found her mom she said, "Mom, I have news!" Her mom said, "What is it, Jane?" Jane said, "I want to stir something up and make it more fun!" \n
\n
Her mom said, "

The twin jumped back in surprise. She had never heard a flower talk before! She asked the flower, "Who are you?"\n
"My name is Pinky," said the flower.\n
\n
The twin was now even more surprised. She asked, "

He asked Jill, "What's in this jar?" Jill smiled and said, "It's sugar! Would you like some?" \n
\n
Jack shook his head and said, "No, thank you. I don't think I should eat sugar. My mom won't allow it." \n
\n
Jill nodded and said, "

Nearby, her mom was watching and called out, "Lucy, come here! What's that you have there?"\n
\n
Lucy proudly held up the hoop and announced, "

Figure 14: Examples of clusters within the TinyStories dataset, discovered by QDG on the TinyStories-33M model. Here we just show samples from four out of 400 `n_clusters`

robust in a reasonably wide region. The envelope slope seems to indicate $\alpha$. We fix $d = 1000$ and $\sigma = 2.0$, sweeping $\alpha = \{0.8, 0.9, 1.0, 1.1, 1.2, 1.3, 1.4, 1.5\}$. For each $\alpha$, we estimate the slope of the envelope. Although there is clear correlation between the estimated envelope and $\alpha$, if we use the envelope slope to estimate $\alpha$, the error is on the order of 0.2, as shown in Figure 17.

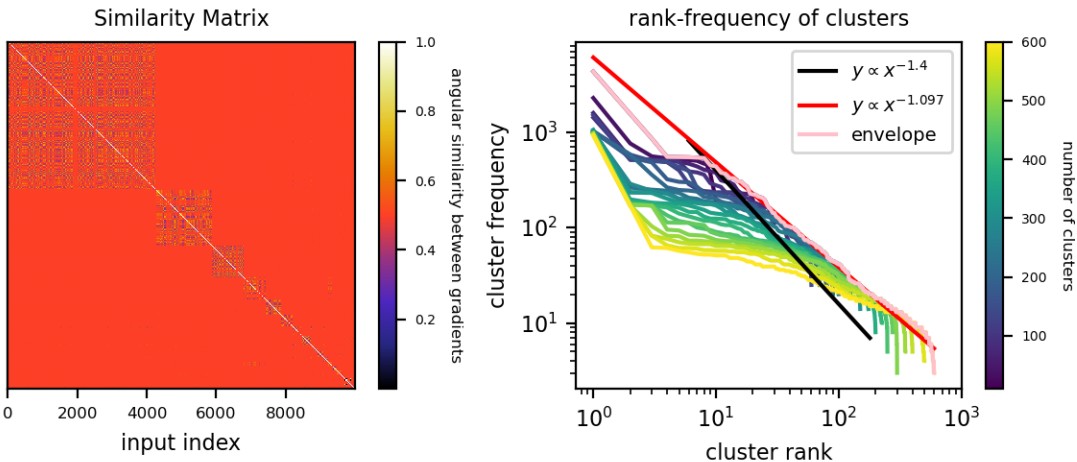

Figure 15: Similarity matrix and rank-frequency plots from QDG on multitask sparse parity. Despite sparse parity having a known decomposition into subtasks which are power law distributed in frequency, we do not recover this same power law from samples. We used $\alpha = 0.4$ for the frequency distribution for an expected rank-frequency power law exponent of -1.4, but measure a rank-frequency envelope slope closer to -1.1.

## F  Parameter and data scaling exponents across studies

In Figure 18, we show $\alpha_N$ and $\alpha_D$ (or possibly $\alpha_S$, depending on the study) for a variety of prior studies of deep learning scaling, as compiled by Villalobos [47]. While the data is messy, it is intriguing that most of the Rosenfeld et al. [2] samples lie below the $\alpha_D = \alpha_N$ line, as our model would predict. The scaling exponents from Hoffmann et al. [7] are slightly closer to our prediction than the relation $\alpha_D = \alpha_N$, which has been proposed by other models of neural scaling laws. Overall though, the existing empirical results are too messy to definitively support or contradict our model.

## G  Estimates of compute used for our experiments

**Multitask sparse parity**: Our training script takes roughly 1-4 hours (depending on network size) to perform a single-epoch training run on a GPU. When training multi-epoch on a fixed dataset, runs take typically between 3-10 minutes, with some outliers taking much longer. Our largest experiment was for Figure 10, where we trained networks of varying width on data with varying distributions over subtasks (with different power law exponents). 467 runs completed with a total running time of approximately 1450 hours. These experiments were run on a cluster with heterogeneous hardware. Availble GPUs include NVIDIA A100, RTXA6000, QUADRORTX6000, GEFORCERTX2080TI, GEFORCERTX2080, GEFORCEGTX1080TI, titan-x, and tesla-v100.

**Pythia model scaling evaluations**: We evaluated Pythia models on NVIDIA A100 80GBs. We do not have available the running time used when computing loss on approximately ten million tokens (for which we reported scaling statistics on), although it was likely less than an hour per model. The most expensive experiments were for Figure 11, where we evaluated the first four models in the Pythia suite across 143 checkpoints for a total of 572 evaluations. We likely used some hundreds of A100-hours for this, though possibly less than 100 hours.

**QDG**: We ran our QDG experiments on an NVIDIA A100 80GB. For the smallest Pythia model and for 10000 samples, it takes a few hours to compute the similarity matrix. We performed this computation only a handful of times.

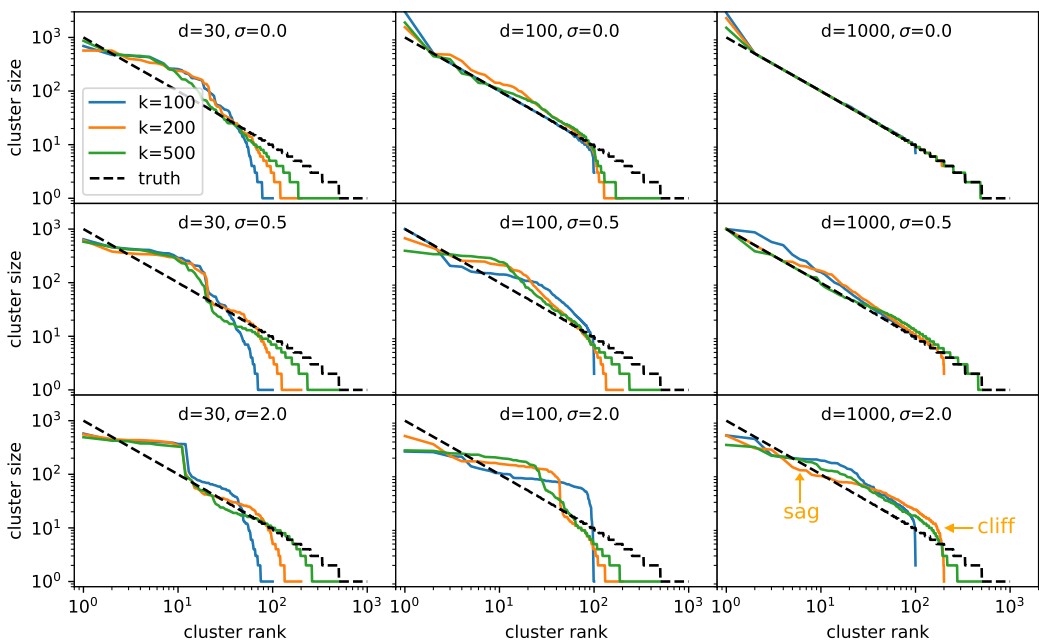

Figure 16: To understand the bias of spectral clustering, we apply spectral clustering to a toy model with different embedding dimension $d$, noise scale $\sigma$ and number of cluster $k$. The high-dimension ($d = 1000$) large-noise ($\sigma = 2.0$) scheme seems to best agree with the LLM results (Figure 5).

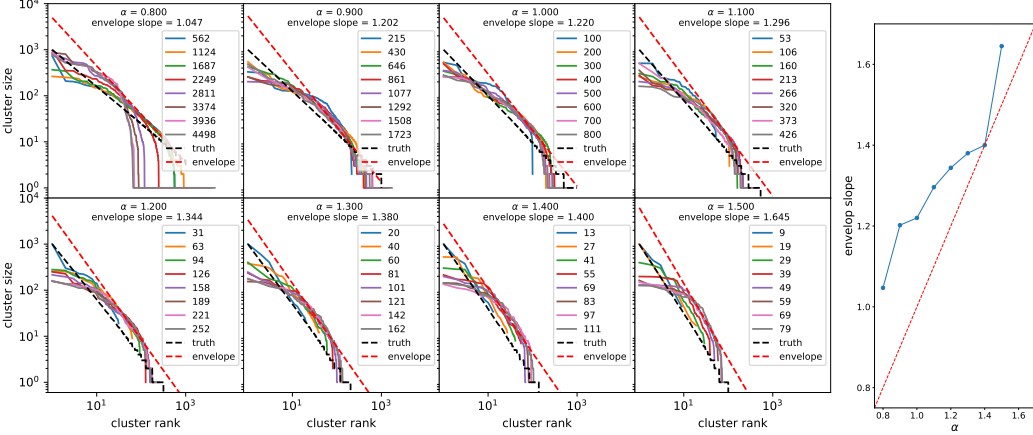

Figure 17: The difficulty of measuring $\alpha$ from curves. We apply spectral clustering to a toy model with different $\alpha$ and number of clusters $k$. For a fixed $\alpha$, different $k$ curves define an envelope. One could use the envelope slope to infer $\alpha$, but this incurs errors around 0.2.

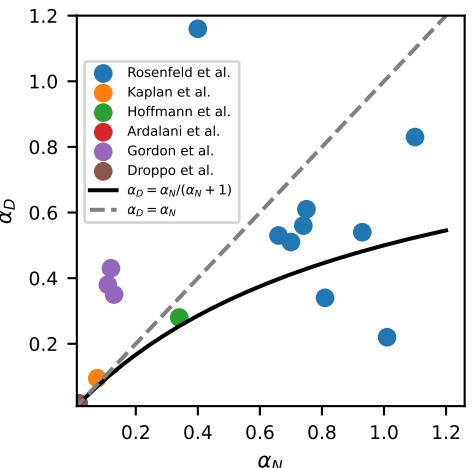

Figure 18: Parameter and data scaling exponents from various studies of deep learning scaling, compiled from the database of neural scaling laws from [47]. Our model of scaling predicts that $\alpha_D = \alpha_N/(\alpha_N + 1)$, indicated with the solid black line. Visible points are from [2, 3, 7, 5, 48]. [49] is above the visible window of the figure.

