# OpenReview forum: "The Quantization Model of Neural Scaling"
_NeurIPS.cc/2023/Conference — NeurIPS 2023 poster_

### Official Review · Reviewer_Kziq · 2023-06-21

**Soundness:** 4 excellent
**Presentation:** 4 excellent
**Contribution:** 3 good
**Rating:** 7
**Confidence:** 4

**Summary:**

This paper proposes a possible mechanism that explains both the phenomenon where the cross-entropy loss of large language models (LLMs) decreases as a power law with respect to the training corpus size, and the phenomenon in which certain capabilities of LLMs emerge spontaneously as the loss becomes low enough. The authors empirically demonstrate that in a toy problem called "multitask sparse parity", in which their assumptions explicitly hold, an MLP network indeed obeys both the scaling laws and the emergence phenomena. Finally, the authors propose an empirical method to auto-discover skills in LLMs and apply this method to provide empirical evidence that their assumptions also hold in LLMs.

**Strengths:**

1. The paper proposes a novel and timely explanation for both the scaling laws phenomenon and the emergence phenomenon in large language models (LLMs).

2. The authors support their explanation with a clear demonstration on a toy task of multitask sparse parity. Essentially, they trained a simple MLP network on multiple sparse parity tasks and demonstrate both the scaling laws phenomenon and the emergence phenomenon when the distribution of the different tasks follows a power law.

3. In addition, the authors demonstrate the relevance of their explanation for LLMs by proposing a novel method for auto-discovering skills in LLMs and show that the discovered skills obey power law frequencies. Using and scaling this method might be of independent interest for both the mechanistic interpretability community and for designing better pre-training datasets.

4. The paper is clearly written and accessible to readers with varying backgrounds.

**Weaknesses:**

There is insufficient evidence to conclusively prove that the Quantization Model accurately depicts the training of large language models.

**Questions:**

Could you provide a practical demonstration of the Quantization Hypothesis in a controlled environment that is neither overly simplistic nor complex, while ensuring a clearer definition of natural language quantas? For instance, perhaps you could utilize a pre-trained language model like GPT-4 to create datasets with well defined quantas in a similar manner to the TinyStories dataset [1].

[1] Ronen Eldan, Yuanzhi Li (2023). TinyStories: How Small Can Language Models Be and Still Speak Coherent English?

**Limitations:**

The authors were very honest in acknowledging less plausible alternative explanations for their empirical findings.

---

> ### Author Rebuttal · Authors · 2023-08-10
>
> Thank you for your feedback and question/suggestion! We agree that more work could be done in evaluating to what extent scaling on real-world datasets satisfies the Quantization Hypothesis.
>
> **Could you provide a practical demonstration of the Quantization Hypothesis in a controlled environment that is neither overly simplistic nor complex, while ensuring a clearer definition of natural language quantas? For instance, perhaps you could utilize a pre-trained language model like GPT-4 to create datasets with well defined quantas in a similar manner to the TinyStories dataset [1].**
>
> This is a cool idea! Based on this suggestion we tried running QDG on the TinyStories-33M model. We get some interesting clusters, although the most interesting TinyStories quanta seem less interesting than the most interesting quanta on The Pile. Some clusters in TinyStories include:
> - Predicting a question mark at the end of some dialogue that asks a question
> - Predicting the correct pronoun based on the name of the subject
> - Predicting quotation marks after ‘said, ‘
> - Predicting a comma after “Suddenly” or “Just then”
> - Predicting a newline after a dedicated space token
>
> Many of the clusters we get seem to reflect very simple repeated patterns in the TinyStories dataset, rather than reflecting sophisticated reasoning. We’ll add examples of some of these clusters to the appendix.

---

> > ### Comment · Reviewer_Kziq · 2023-08-20
> > **Thank you**
> >
> > Thank you for the thorough rebuttal and additional experiment you performed. After reading the other reviews, I have decided to keep my score unchanged.

---

### Official Review · Reviewer_S4dM · 2023-07-03

**Soundness:** 2 fair
**Presentation:** 2 fair
**Contribution:** 3 good
**Rating:** 5
**Confidence:** 4

**Summary:**

This paper proposes a hypothesis that there exists a universal and crucial discrete set of computations (quanta) for reducing loss in various prediction problems, and the model's performance is determined by successfully learning these computations. Through this hypothesis, it demonstrates the relationship between power law neural scaling and the effective learning of more quanta, which cannot be solved solely by the memorization ability of the existing model, particularly in complex problems. Additionally, the paper proposes "Quanta Discovery from Gradients" as a method to auto-discover quanta.

**Strengths:**

1. Based on the assumption that there exists a discrete set of computations, known as "quanta," which determines the performance of the model, the paper provides hints for understanding the emergent behavior observed in the scaling law of the model.
2. The objective of the study is to interpret power law scaling and emergent behavior phenomena by using a multitask sparse parity dataset that cannot be solved solely by the model's memorization ability.
3. Through experiments conducted on the multitask sparse parity dataset, the paper shows that when the frequency of using quanta follows a power law distribution, power law neural scaling can occur as the model learns more quanta.

**Weaknesses:**

1. The paper lacks a clear definition of quanta, which could benefit from further elaboration and clarification.
2. Insufficient explanation is provided regarding the criteria used to determine quanta through the Quanta Discovery from Gradients (QDG) method.
3. Extending the experiments on the Quantization Hypotheses, using the proposed toy dataset, to Language Models (LLM) is hindered by the lack of clarity regarding the relationship between LLM tokens and quanta.
4. Section 5 claims that "Clustering based on inputs or outputs therefore seems unlikely to discover quanta in language modeling," but lacks substantial empirical evidence to support this statement.
5. The explanation concerning the relationship between quanta and gradients, which forms the basis for determining quanta in the QDG method, needs to be further developed and elaborated upon.

**Questions:**

1. In Figure 1, what decoding method was used for the auto-discovered quanta?
2. The experiments on the toy dataset appear to be influenced by the decoding algorithm (such as greedy, beam search, random). Is there any consideration given to this aspect?
3. In Figure 4, what is the relationship between the highlighted red section in the prompt and each graph?
4. In Section 5, it would be beneficial to include details about the dimensional of the gradients used in the Quanta Discovery from Gradients (QDG) method. The absence of explicit notation indicating the shape of each symbol for explaining QDG makes it challenging to ascertain the computational complexity involved in implementing QDG.

**Limitations:**

There is a need to demonstrate the effectiveness of the proposed Quanta Discovery from Gradients (QDG) method in identifying quanta in various Language Models (LLM) and datasets.

---

> ### Author Rebuttal · Authors · 2023-08-10
>
> Thank you for your detailed feedback and questions, and for highlighting the importance of clarifying our definitions in the paper! We think the clarifications we’ll make, prompted by your review, will significantly improve the paper. We’ll respond point by point:
>
> ### Weaknesses:
> **1. The paper lacks a clear definition of quanta, which could benefit from further elaboration and clarification.**
>
> Here is the definition of quanta which we’ll add to the paper:
>
> _Definition of quantum (plural quanta)_: An indivisible computational module that, for example, retrieves a fact or implements an algorithm.
>
> Although quanta cannot be learned instantaneously in practice, we expect their formation to correspond with a quick drop in loss akin to an upside-down sigmoid. This was observed for the formation of “induction heads” in Olsson et al. (2022), “In-context Learning and Induction Heads”. We interpret the induction circuit described in that paper as one such quantum, and propose that LLMs can be understood as an ensemble of such modules.
>
> **2. Insufficient explanation is provided regarding the criteria used to determine quanta through the Quanta Discovery from Gradients (QDG) method.**
>
> Thanks again for pushing us toward a clearer presentation!
>
> _QDG_: Operationally, we identify quanta as clusters of next-token prediction samples (found by spectral clustering of gradients) that pass subsequent vetting for task coherence.
>
> While QDG does not on its own produce a mechanistic understanding of how the quanta work, it does suggest the function that these modules have. For instance, in one QDG cluster, all samples involve predicting a number to continue a numerical sequence, and we think that this suggests that there is a module in the network that is responsible for this capability/behavior.
>
> **3. Extending the experiments on the Quantization Hypotheses, using the proposed toy dataset, to Language Models (LLM) is hindered by the lack of clarity regarding the relationship between LLM tokens and quanta.**
>
> The quantity being measured in LLM scaling laws is the mean cross-entropy loss for next-token prediction in the training distribution of text. In our model, a quantum in an LLM is a computational module which improves the network’s ability to do next-token prediction on some fraction of tokens in the training distribution of text. To recover LLM scaling laws, we claim that these fractions (frequencies at which the quanta are used) follow a power law.
>
> **4. Section 5 claims that "Clustering based on inputs or outputs therefore seems unlikely to discover quanta in language modeling," but lacks substantial empirical evidence to support this statement.**
>
> When we tried to cluster next-token prediction samples just based on what the next token was, or based on the last tokens of the input, we didn’t get interesting clusters. This is a minor point, but admittedly not well supported by any experiments we showed in the paper. We’ll either remove this claim or support it with experiments.
>
> **5. The explanation concerning the relationship between quanta and gradients, ..., needs to be further developed and elaborated upon.**
>
> When we clustered samples according to gradients, we intended that this would cluster them according to whether the model used similar knowledge or circuitry to perform prediction on those samples. Consider some quantum (module) in the network. If it is important for prediction on a pair of samples, then the subset of network parameters that are part of the module will perhaps have nonzero and similar gradients for those samples. If on a different sample that module is irrelevant to prediction, then the module’s parameters might have gradients of close to zero (or at least different gradients from those samples which relied on the module). Therefore gradients on samples which rely on the same computational module will have higher cosine similarity than between samples which don’t rely on the same module. Thank you for pointing out that we didn’t justify this at all in the current version of the paper! We’ll add some text like this motivating the use of gradients to identify quanta in Section 5 when we introduce QDG.
>
> ### Questions
> **1. In Figure 1, what decoding method was used for the auto-discovered quanta?**
>
> With quanta, we’re interested in the capabilities present in a single forward pass of a model. The samples shown in Figure 1 are next-token prediction samples, where the token to be predicted is highlighted in red. We only applied clustering to samples (where a sample involves predicting a single token from its context) if the model’s cross-entropy loss on them was low (we chose a threshold of 0.1 nats). These are samples where the model would correctly predict the next token if greedy decoding was used.
>
> **2. The experiments on the toy dataset appear to be influenced by the decoding algorithm (such as greedy, beam search, random). Is there any consideration given to this aspect?**
>
> The multitask sparse parity task is a supervised binary classification problem which we train MLPs to solve, so we don’t think that decoding is relevant. Categorical cross-entropy loss seems like a natural metric of model performance on this task.
>
> **3. In Figure 4, what is the relationship between the highlighted red section in the prompt and each graph?**
>
> In Figure 1 and Figure 4, the highlighted red text indicates the token that the LM was predicting from the text shown before it. So in Figure 4, we show how cross-entropy loss changes with model scale in predicting the highlighted red token from its context. These examples are from the test set of The Pile corpus.
>
> **4. …The absence of explicit notation indicating the shape of each symbol for explaining QDG makes it challenging to ascertain the computational complexity involved in implementing QDG.**
>
> Good point, sorry for the omission! We’ll add details about the dimensionality of the model and the complexity of running QDG to Section 5.

---

> > ### Comment · Reviewer_S4dM · 2023-08-16
> >
> > Thank you for the detailed answers and results.
> >
> > Some of my concerns have been addressed, but i still my concern there is insufficient evidence to support the hypotheses about the proposed quantization model showing consistent results in large-scale language models. However, I would like to raise the score considering the potential demonstrated by the Quantization Model to understand "neural scaling" and the accompanying experiments that provide support for this.

---

### Official Review · Reviewer_7dsB · 2023-07-06

**Soundness:** 3 good
**Presentation:** 4 excellent
**Contribution:** 4 excellent
**Rating:** 8
**Confidence:** 4

**Summary:**

This paper proposes a new way of understanding scaling laws. Namely, it shows that capabilities can be thought of as being broken into discrete units (quanta) which themselves follow a power law and have an inherent ordering--called the Q Sequence. This combined with the fact that 1) an increasing subset of quanta are learned over various scales and 2) model and data sizes can be related to the number of quanta gives rise to the scaling laws shown in previous works. The paper also proposes a method to cluster examples by unit-normalized gradient, and empirically confirms within margin of error that theory agrees with observed data. The quantization model explains a sudden emergence of new capabilities with scale: certain quanta must be learned before the model can carry out certain tasks.

**Strengths:**

Simple, effective model to understand scaling laws

**Weaknesses:**

While the paper shows that the model scaling trend agrees with theory on real data, it does not empirically validate whether the same alpha of 0.083 matches the data scaling theory. Also, it would be interesting to show this holds on non LLM tasks like image classification, since the (titular) claim is "neural scaling".

**Questions:**

Everything is clear.

**Limitations:**

Not that I am aware of

---

> ### Author Rebuttal · Authors · 2023-08-10
>
> Thank you for your feedback. We’re glad you found the paper compelling!
>
> **While the paper shows that the model scaling trend agrees with theory on real data, it does not empirically validate whether the same alpha of 0.083 matches the data scaling theory. Also, it would be interesting to show this holds on non LLM tasks like image classification, since the (titular) claim is "neural scaling".**
>
> It would indeed be great to evaluate whether the Quantization Model holds on other types of data, e.g. for vision tasks like you suggest. Perhaps this could be done in future work.
>
> For data scaling, Figure 11 and Figure 17 of the Appendix (Supplementary Material) may partially address this. For the Pythia LLM scaling suite, all models are trained on the same amount of data, so we can’t study multi-epoch data scaling. However we do have intermediate training checkpoints for these models, so we can study scaling in training steps. In Figure 17 we see that the training curve is not obviously a clean power law, so it is not clear where to fit a line to the curve. For the lines we do fit, we get a slope which is less negative than -0.083, which is what we’d expect from our theory, although the measured slopes of between -0.037 and -0.06 are shallower than we’d expect from theory: -0.083 / (1 + 0.083) = -0.077. This could be due to a suboptimal choice of hyperparameters when EleutherAI trained these models. Figure 17 shows empirical scaling exponents in parameters and data from a variety of scaling papers. The most reliable point is probably the green point from the Chinchilla paper (Hoffmann et al.), which lies slightly above our curve $\alpha_D = \alpha_N / (\alpha_N + 1)$.

---

### Official Review · Reviewer_2SwD · 2023-07-10

**Soundness:** 3 good
**Presentation:** 3 good
**Contribution:** 3 good
**Rating:** 5
**Confidence:** 3

**Summary:**

The authors investigate neural scaling laws and propose an explanation for the power law scaling that is observed in addition to the emergence of new behaviours.

**Strengths:**

The ideas are interesting and I found some of the experiments such as per token losses on the language models to be quite interesting.



**Weaknesses:**

I found the presentation of Section 5 to be a bit confusing.

I also found the connection to emergent behaviours to be a bit speculative. Specifically I think the idea of looking at per token losses/etc to be quite interesting, but I wouldn't necessarily say that I am convinced that this is emergence in the same way I'd think of emergence on some BigBench tasks, for example.

**Questions:**

Could you elaborate on why you feel that this demonstrates emergent capabilities in language models?

You mention the spectral clustering hyperparameters: how sensitive are other parts of this work to broad choices of hyperparameters? For example how are you setting batch size / LR for the LM experiments?

In Figure 5 right panel there is some interesting shape in the yellow curves-- I'm curious what you think of those?


**Limitations:**

Yes, the limitations are mostly well addressed.

---

> ### Author Rebuttal · Authors · 2023-08-10
>
> Thank you for your feedback and questions! We’re glad you found our ideas and some of our experiments interesting. Before responding to your questions point by point, we’d like to apologize for not making our most interesting contribution more clear, namely our notion of quantization and of quanta. Essentially we argue that large neural networks are implicitly ensembles of indivisible computational modules (quanta). We view our work as a step toward validating Marvin Minsky’s Society of Mind theory, that intelligent systems can be thought of as collections of many smaller systems performing specialized jobs. Intriguingly, we were able to reconcile this view of neural networks with scaling laws while also providing a framework for understanding emergence, although understanding emergence is a less important point of the paper. We will edit the paper to better emphasize our notion of quantization and its relation to Minsky.
>
>
> **I found the presentation of Section 5 to be a bit confusing.**
>
> We’re sorry about this! Section 5 could definitely be better motivated and explained. The first paragraph in particular jumps into some confusing discussion about partitioning the problem of language modeling into subtasks without motivating that. We will edit the paper to improve the clarity of this section. In Section 5, we were ultimately interested in trying to discover what some of the computational modules (quanta) in LLMs are.
>
> **I also found the connection to emergent behaviours to be a bit speculative. Specifically I think the idea of looking at per token losses/etc to be quite interesting, but I wouldn't necessarily say that I am convinced that this is emergence in the same way I'd think of emergence on some BigBench tasks, for example.**
>
> One advantage of looking at per-token loss is that it avoids some of the critiques of emergent abilities brought up by Schaeffer et al. (2023) “Are Emergent Abilities of Large Language Models a Mirage?” -- some examples of emergence in BigBench may be due to the choice of metric. In contrast, the loss on a single token is a smooth function of the model’s output, so if we see a sharp drop in loss, this suggests that there is some genuine qualitative difference in the model.
>
> **Could you elaborate on why you feel that this demonstrates emergent capabilities in language models?**
>
> One of the core questions we’re interested in with this paper is: how does scaling change what neural networks learn? Our hypothesis is that scaling adds new computational modules (quanta) to the network that weren’t present before.
>
> One prominent prior model of neural scaling is Sharma and Kaplan (2022) “Scaling Laws from the Data Manifold Dimension”. In this model, neural networks are understood as approximating a function defined on some “data manifold”, and the effect of scaling is to smoothly increase the resolution/precision at which this function is approximated. This can explain power law scaling, but there seems to be some tension between this view, where scaling leads to a smooth change in what models learn, and the sharp improvements with scale we see in network performance on some tasks (e.g. BigBench tasks with high “breakthroughness”). Now, one way of resolving this tension is to argue that emergence is an artifact of one’s choice of metric like Schaeffer et al. (2023) do. But we propose an alternative view with our model, where smooth power law scaling in mean loss averages over many discrete changes as indivisible modules (quanta) are added to the model.
>
> In the multitask sparse parity dataset we construct, this model of scaling holds -- emergence is real: performance on subsets of the data sharply transitions from random-guess performance to perfect performance with increasing scale. But we still get smooth power laws in mean loss. So a very strong notion of emergence can occur for data with the right structure.
>
> If this story described language modeling, then we could understand emergent abilities in language models as the result of new quanta being learned. If a benchmark exhibits emergence, we’d say that the quanta relevant to solving that task are learned at a similar scale. At that scale, models transition to good performance on the task since the necessary quanta are now present, and weren’t present at smaller scale.
>
> **You mention the spectral clustering hyperparameters: how sensitive are other parts of this work to broad choices of hyperparameters? For example how are you setting batch size / LR for the LM experiments?**
>
> For the LM experiments, we didn’t train any LMs ourselves but instead just used the open source Pythia suite from EleutherAI. For the multitask sparse parity experiments, we just used a large batch size and an AdamW learning rate of 1e-3 that worked well in practice. We’d be happy to run some grid searches over batch size and LR and include these results in the appendix!
>
> **In Figure 5 right panel there is some interesting shape in the yellow curves-- I'm curious what you think of those?**
>
> We’re not quite sure what’s causing the deviations from a clean power law that we see in the rank-frequency curves in Figure 5. In Appendix D.2 we experimented with a toy model of clustering and found that we could get similar looking curves when the dimension is high and when the noise was high -- see the lower right panel of Figure 15 if interested. We will edit the paper to point this out and refer readers to the appendix.

---

> > ### Comment · Reviewer_2SwD · 2023-08-16
> >
> > Thanks for the response, I'm happy to raise my score to a 5.

---

### Official Review · Reviewer_7ZcN · 2023-07-13

**Soundness:** 4 excellent
**Presentation:** 4 excellent
**Contribution:** 4 excellent
**Rating:** 8
**Confidence:** 4

**Summary:**

The paper proposes that the capabilities of a neural network are composed of discrete “quanta” that each help reduce loss on a subset of training examples, and that are learned in approximately decreasing order of frequency. If the quanta are present in a Zipfian (power-law) frequency distribution in the training data, and each quantum is approximately equally valuable in the examples where it’s present, such a theory would provide a mechanistic explanation for power law scaling of loss with respect to model and dataset size.

First the authors validate this hypothesis on a toy dataset constructed to have this quantized structure (by being composed of several independent tasks each with a different prefix key). Then they find similar patterns in real language models, and use an algorithm they call “quanta discovery with gradients” to identify potential quanta in these models and their data.

Their results are supportive of the quantization hypothesis, but with a high degree of uncertainty.

**Strengths:**

The hypothesis is a beautiful and important claim if true, and the paper provides a well presented and solid chunk of evidence that it is. The toy dataset experiment is simple and straightforward and demonstrates that the quantization hypothesis works under ideal conditions, and the LM experiment shows that similar phenomena are also present in real models, while proposing a reasonable initial approach to quanta discovery. The results on monogenic vs. polygenic samples also point pretty clearly towards the validity of quantization.

Given QDG or any other model of what specifically the quanta might be, the hypothesis also enables testable predictions about scaling laws.


**Weaknesses:**

The multitask parity setting seems too obviously likely to lead to quanta, so it doesn’t seem to prove very much (although it’s useful to set up the framing, almost like a thought experiment).

The QDG methodology is also a bit disappointing in a few ways. By clustering the gradients, it assumes that quanta (conceptually defined without reference to model structure) will be localized in the model—which is probably true, but might otherwise have been a testable hypothesis. It’s also too slow for the authors to have applied it to models larger than the smallest Pythia LM, greatly limiting its applicability.


**Questions:**

How much do you believe the hypothesis overall? Might there also be a significant role for “interaction terms” between quanta or are they close to being fully independent?

Do you have early ideas about alternatives to QDG, especially alternatives that might be more computationally tractable?


**Limitations:**

The authors acknowledge the limitations of their methods and experiments.

---

> ### Author Rebuttal · Authors · 2023-08-10
>
> Thank you for your constructive feedback and questions! We’re glad you found the paper to be interesting and valuable. We’ll comment on the weaknesses and respond to your questions below:
>
> **The multitask parity setting seems too obviously likely to lead to quanta, so it doesn’t seem to prove very much (although it’s useful to set up the framing, almost like a thought experiment).**
>
>
> We agree that the multitask sparse parity dataset is a bit contrived. One of the ways in which it’s valuable is that it shows that the mechanism of power law neural scaling depends on the structure of the data. For instance, in Sharma and Kaplan (2022) “Scaling Laws from the Data Manifold Dimension”, they train neural networks on a set of toy regression problems of varying input dimension and observe power law scaling where the scaling exponent is determined by the dimension of the space that the function is defined on. For those sorts of tasks, their model of neural scaling, which views neural networks as approximating a function defined on some manifold with better and better precision with increasing scale, seems to hold. But our experiments on multitask sparse parity show that one can also get power law scaling in accordance with the Quantization Model when the data is fundamentally discrete, and where neural networks learn an increasing number of discrete computations with increasing scale, rather than globally approximating a function on a manifold with higher precision. So the mechanism of power law scaling depends on the structure of the data. Now when we observe power law scaling in the wild, we can ask whether the Sharma and Kaplan model of scaling or the Quantization Model of scaling, or maybe something else, best describes what’s going on.
>
>
> **The QDG methodology is also a bit disappointing in a few ways. By clustering the gradients, it assumes that quanta (conceptually defined without reference to model structure) will be localized in the model—which is probably true, but might otherwise have been a testable hypothesis. It’s also too slow for the authors to have applied it to models larger than the smallest Pythia LM, greatly limiting its applicability.**
>
> Since we cluster using the (almost) full model gradients (across all layers of the network) our method might still work even if computations are spread across large parts of the model. We just assume that gradients will have some consistency among samples where prediction relies on the same pieces of knowledge/computation. We hope that more principled and efficient methods could be developed in future work.
>
>
> **How much do you believe the hypothesis overall? Might there also be a significant role for “interaction terms” between quanta or are they close to being fully independent?**
>
> Indeed it might be unrealistic to think of the quanta as being fully independent in language modeling -- great point! In the current draft of the paper, when we talk about samples being polygenic, we imagine something like different circuits pushing the logits in a good direction independently. But realistically, polygenicity is probably more complicated than this. On some samples, a model’s computation might be more integrated, where loss is only lowered if multiple quanta are present simultaneously. In the final draft, we will clarify that polygenicity can be complicated, and the quanta might not lower the loss entirely independently of each other.
>
>
> One interesting biological analogy is the fact that in humans, blue eyes vs. brown eyes is a monogenic trait. However, this gene only matters in the context of many other genes, e.g. the genes that are responsible for humans having eyes in the first place! So it’s sort of implicitly polygenic. Many quanta in LLMs could be similar. A quantum might depend on other quanta, and so would be polygenic in some sense, but deleting it would sharply remove some capability of the model.
>
>
> **Do you have early ideas about alternatives to QDG, especially alternatives that might be more computationally tractable?**
>
> Using activations instead of gradients could lower the dimension considerably. Also using random projections of the gradient instead of the full gradients approximately preserves cosine similarity via the Johnson-Lindenstrauss lemma. Furthermore, the idea of clustering samples to discover quanta only makes sense when samples are monogenic. Ultimately, we’d like a scheme which enumerates the quanta and then for a given sample identifies which quanta (possibly many of them) were relevant for prediction on that sample.

---

> > ### Comment · Reviewer_7ZcN · 2023-08-19
> >
> > Thanks for the follow-up! I'm glad to see you mentioned adding TinyStories quanta to the appendix in a different rebuttal; I should have suggested something like that too!
> >
> > I think your framing in the first question answer (regarding the Sharma/Kaplan and quantization models) is clearer and more explicit motivation for the multitask parity task than I remember seeing in the paper; could be worth making the point equally directly there.

---

### Decision · Program_Chairs · 2023-09-21

**Decision:**

Accept (poster)

**Comment:**

In this paper, the authors develop a model of neural scaling where smooth scaling laws average over small emergent changes in model performance on subtasks denoted as "Quantization Model of Neural Scaling Laws". This is a novel theoretical model of understanding scaling laws which have verifiable predictions in practical LLMs.

Reviewers pointed out some of strengths and weakness as:

Strength
- novel and timely explanations for scaling laws and emergence in large language models
- compelling case by first clearly demonstrating their theory on a toy model
- provides results showing theory's applicability to real-world LLMs
- the hypothesis also enables testable predictions about scaling laws.
- simple, effective model to understand scaling laws
- paper is clearly written and accessible to readers with varying backgrounds.


Weakness
- quanta defined in the paper is difficult to identify in practical LLMs and too slow to scale
- results are supportive of the quantization hypothesis, but with a high degree of uncertainty.
- multitask parity setting not sufficient for validation of hypothesis
- connection to emergent behaviors to be a bit speculative
- insufficient evidence to support the hypotheses about the proposed quantization model showing consistent results in large-scale language models

Reviewers highlighted that "the paper's novel contributions and rigorous approach warrant publication". "It will spark valuable discussion and inspire further research on these topics".  All the reviewers recommended accepting the paper (8, 8, 7, 5, 5, showing varying degree of support) saying that the paper is technically strong with novel ideas and excellent impact. While there are some concerns on speculative nature and how properly the proposed model explains scaling law and emergent behavior of LLM, it definitely provides a step towards providing theory for these phenomena and is worth sharing with the broad NeurIPS audience.